# Microbiome analysis of 940 lung cancers in never-smokers reveals lack of clinically relevant associations

In spite of the growing interest in the microbiome in human cancer, there are currently only small-scale lung cancer microbiome studies conducted directly on tissue. As part of the Sherlock-*Lung* study, we studied the microbiomes of 940 lung cancers (4090 samples) in never smokers (LCINS) directly from lung tissue using three data types: 16S rRNA gene sequencing (16S), whole-genome sequencing (WGS) with paired blood, and RNA-seq. We observe very low biomass and few microbiome associations in LCINS using 16S and WGS tissue. Using RNA-seq, we observe more total microbial reads, and decreased relative abundance of several commensal bacteria at the genus and species levels in tumors relative to paired normal lung tissue. Among all datasets, we see no consistent associations between the lung tissue microbiome, or circulating bacterial DNA, and any available demographic and clinical features, including age, sex, genetic ancestry, second-hand tobacco smoking exposure, LCINS histology, stage, and overall survival. We also observe no microbiome associations with any human genomic alterations within the same samples. Every null result should be interpreted with caution given the possibility of future methodological breakthroughs. However, all together, using multiple data types in nearly 1000 patients, we find no substantive role for the lung cancer microbiome in treatment-naïve LCINS.

The human cancer microbiome is a rapidly growing field of research. To date, most major studies on the human cancer microbiome have focused on organs with high bacterial abundance, e.g., mouth, stomach, and colon, identifying connections between the microbiome and cancer incidence or progression. Additionally, several specific microbes have been shown to produce genotoxins, suggesting a possible role in cancer initiation. These include *Helicobacter pylori*, a Group 1 carcinogen which causes stomach cancer[1], as well as pks+ *Escherichia coli*[2–10], *Bacteroides fragilis*[11–15], and *Fusobacterium nucleatum*[16–22], each associated with colorectal cancer. Resultantly, enthusiasm for the microbiome as a target for cancer early detection[23–25], prevention, and treatment[26] has grown significantly in recent years. Despite this, research on the cancer microbiomes of most organs has been limited, including on the lung.

Relatively little is known about the lung microbiome, even in healthy individuals. Historically, the lungs have been considered sterile organs due to repeated failure to culture bacteria from lung samples[27]. This idea has since been challenged using culture-free sequencing methods. Much of the current research on the lung microbiome is derived from samples collected via sputum or bronchoalveolar lavage (BAL). Studies performed on healthy individuals and cancer patients with samples collected using BAL have characterized the lung microbiome as being similar in composition to the oral and upper airway microbiomes, albeit at much lower total abundance[28–32]. In contrast, small-scale studies conducted on surgically removed tumor and normal lung tissue–which theoretically precludes contamination from the upper airways[33]–identified much lower proportions of upper airway bacteria[34–38]. A recent study of the murine lung microbiome concluded

✉e-mail: landim@nih.gov

**Table 1 | Demographic and clinical features of study subjects**

| Characteristic | 16S N = 701[a] | RNA N = 661[a] | WGS N = 811[a] |
|---|---|---|---|
| Age At Diagnosis | 64 (57, 72) | 64 (57, 72) | 65 (58, 72) |
| Unknown | 0 | 0 | 4 |
| Sex | | | |
| Female | 563 (80%) | 529 (80%) | 639 (79%) |
| Male | 138 (20%) | 132 (20%) | 172 (21%) |
| Ancestry | | | |
| African (AFR) | 2 (0.3%) | 0 (0%) | 4 (0.5%) |
| American (AMR) or Mixed | 38 (5.4%) | 39 (5.9%) | 28 (3.5%) |
| East Asian (EAS) | 388 (55%) | 354 (54%) | 338 (42%) |
| European (EUR) | 272 (39%) | 267 (40%) | 441 (54%) |
| Unknown | 1 | 1 | 0 |
| Study Site | | | |
| Connecticut, USA | 0 (0%) | 0 (0%) | 22 (2.7%) |
| Florida, USA | 0 (0%) | 0 (0%) | 11 (1.4%) |
| Hong Kong | 132 (19%) | 130 (20%) | 113 (14%) |
| IARC (Serbia, Czech Republic, Romania, Poland, Russia) | 204 (29%) | 195 (30%) | 190 (23%) |
| Lima, Peru | 4 (0.6%) | 4 (0.6%) | 0 (0%) |
| Massachusetts, USA | 0 (0%) | 0 (0%) | 26 (3.2%) |
| Mexico City, Mexico | 14 (2.0%) | 16 (2.4%) | 0 (0%) |
| Minnesota, USA | 12 (1.7%) | 13 (2.0%) | 13 (1.6%) |
| New York, USA | 12 (1.7%) | 11 (1.7%) | 13 (1.6%) |
| Nice, France | 26 (3.7%) | 44 (6.7%) | 53 (6.5%) |
| Quebec, Canada | 0 (0%) | 0 (0%) | 113 (14%) |
| Taiwan | 218 (31%) | 202 (31%) | 184 (23%) |
| Toronto, Canada | 71 (10%) | 39 (5.9%) | 68 (8.4%) |
| Valencia, Spain | 8 (1.1%) | 7 (1.1%) | 5 (0.6%) |
| Stage | | | |
| I | 407 (61%) | 368 (59%) | 496 (64%) |
| II | 115 (17%) | 113 (18%) | 131 (17%) |
| III | 108 (16%) | 107 (17%) | 120 (15%) |
| IV | 38 (5.7%) | 39 (6.2%) | 30 (3.9%) |
| Unknown | 33 | 34 | 34 |
| Histology | | | |
| Adenocarcinoma | 621 (89%) | 584 (88%) | 695 (86%) |
| Adenosquamous carcinoma | 12 (1.7%) | 9 (1.4%) | 11 (1.4%) |
| Carcinoid tumor | 23 (3.3%) | 23 (3.5%) | 58 (7.2%) |
| Squamous cell carcinoma | 36 (5.1%) | 36 (5.4%) | 34 (4.2%) |
| Other | 9 (1.3%) | 9 (1.4%) | 13 (1.6%) |

[a]Median (Q1, Q3); n (%)

that although both methods may be valid for studying the lung microbiome, samples collected from BAL fluid versus directly from lung tissue within the same animals can be distinguished via beta diversity analysis[39].

Alterations in the lung microbiome are connected with several diseases[40], such as chronic obstructive pulmonary disease[41–44], asthma[45–47], and idiopathic pulmonary fibrosis[48]. Furthermore, changes in the lung microbiome of mice have been shown to influence the development of multiple sclerosis in the brain[49]. Many studies have also identified differences in the lung microbiomes of healthy versus cancer patients[32,36–38,50–54] and tumor versus adjacent normal tissue[34,36–38], and several have found associations with tumor clinical features, such as histology[55], stage[34], and progression[56]. However, these studies are predominantly based on small sets of patients (on average, fewer than 100 subjects, ranging from 10[51] to 176[37] subjects

total), resulting in discrepant results. Additionally, most datasets are composed primarily of smokers, and thus, the role of the microbiome specifically in never-smoker lung cancer is largely unstudied.

In this study, we used 16S sequencing to analyze the microbiome of 701 surgically removed treatment-naive lung cancers in never smokers (LCINS) plus 563 tumor-adjacent normal lung samples, the largest sample collection to date. To further increase the size of our study, we leveraged an additional 1623 WGS samples (tumor, normal lung, blood) and 1203 RNA-seq samples (tumor, normal lung) collected as part of Sherlock-*Lung* and investigated bacterial reads in these samples. With considerable overlap of subjects between datasets, this study includes a total of 4090 samples from 940 cancer patients who were treatment naive at the time of sample collection. Despite the comprehensive analysis, we found no evidence for clinically relevant associations between the composition or diversity of the lung cancer microbiome and LCINS demographics, tumor characteristics, previous respiratory diseases, genomic features, and survival or recurrence.

## Results
### Description of study samples
This study is based on the Sherlock-*Lung* project[57] of LCINS. Briefly, as part of Sherlock-*Lung* (hereafter referred to simply as Sherlock), we have analyzed WGS[58,59], 16S, and RNA-seq data from hundreds of LCINS across North and South America, Europe, and Asia, together with epidemiological, clinical, and morphological features.

Specifically, we examined the microbiomes of 940 LCINS patients, 740 females and 200 males of median age 64.7 years, with 639 paired adjacent normal tissue plus 447 WGS blood samples (Table 1 and Supplementary Data 1). Sex was self-reported and confirmed via WGS where available. Based on WGS-derived genetic ancestry, this cohort includes 441 patients of European ancestry from the United States, Canada, and Europe; 338 of East Asian ancestry from Hong Kong, Taiwan, the United States, and Canada; 28 of Native American/Mixed ancestry from Europe and Canada, plus 4 of African ancestry from the United States and Canada (Table 1 and Supplementary Data 1). For patients without WGS data, ancestry was self-reported, including 58 patients of East Asian ancestry from Hong Kong, Taiwan, and Canada; 46 of European ancestry from Europe and the United States; and 24 of Native American/Mixed ancestry (Table 1 and Supplementary Data 1). One patient from Canada was of unknown ancestry.

As is typical in LCINS, the most common histology was adeno-carcinomas (n = 811), followed by carcinoid tumors (n = 60), squamous cell carcinomas (n = 40), and various other tumor types (n = 29) (Table 1 and Supplementary Data 1). The majority of tumors (n = 522) and normal lung tissue (n = 278) were sequenced using all three approaches: WGS, 16S, and RNA-seq (Fig. 1a and Supplementary Data 1).

### Multi-omic identification of bacterial reads
Recently, debate has emerged about best practices for microbiome research[23,60–62] using next-generation sequencing (NGS) after several methodological errors were identified in a major pan-cancer study on the cancer microbiome[60]. These errors resulted in millions of una-ligned human sequences being misidentified as bacterial, which affected some of the findings of the original paper[62]. To avoid assigning human reads to bacterial genomes, as discussed in Gihawi et al.[61], we aligned all reads to the CHM13 T2T genome[63] to filter out as many human sequences as possible prior to taxonomic assign-ment with Kraken2[64] (Fig. 1b), then extracted unaligned reads from this realignment for use with Kraken2. Following taxonomic assign-ment, we used Bracken[65] to adjust read counts at the genus level for both WGS and 16S sequencing, but chose not to use Bracken for the RNA-seq dataset as Bracken was developed for DNA-based sequen-cing ("Methods"). Taxonomic assignment results are presented in Supplementary Data 2–4.

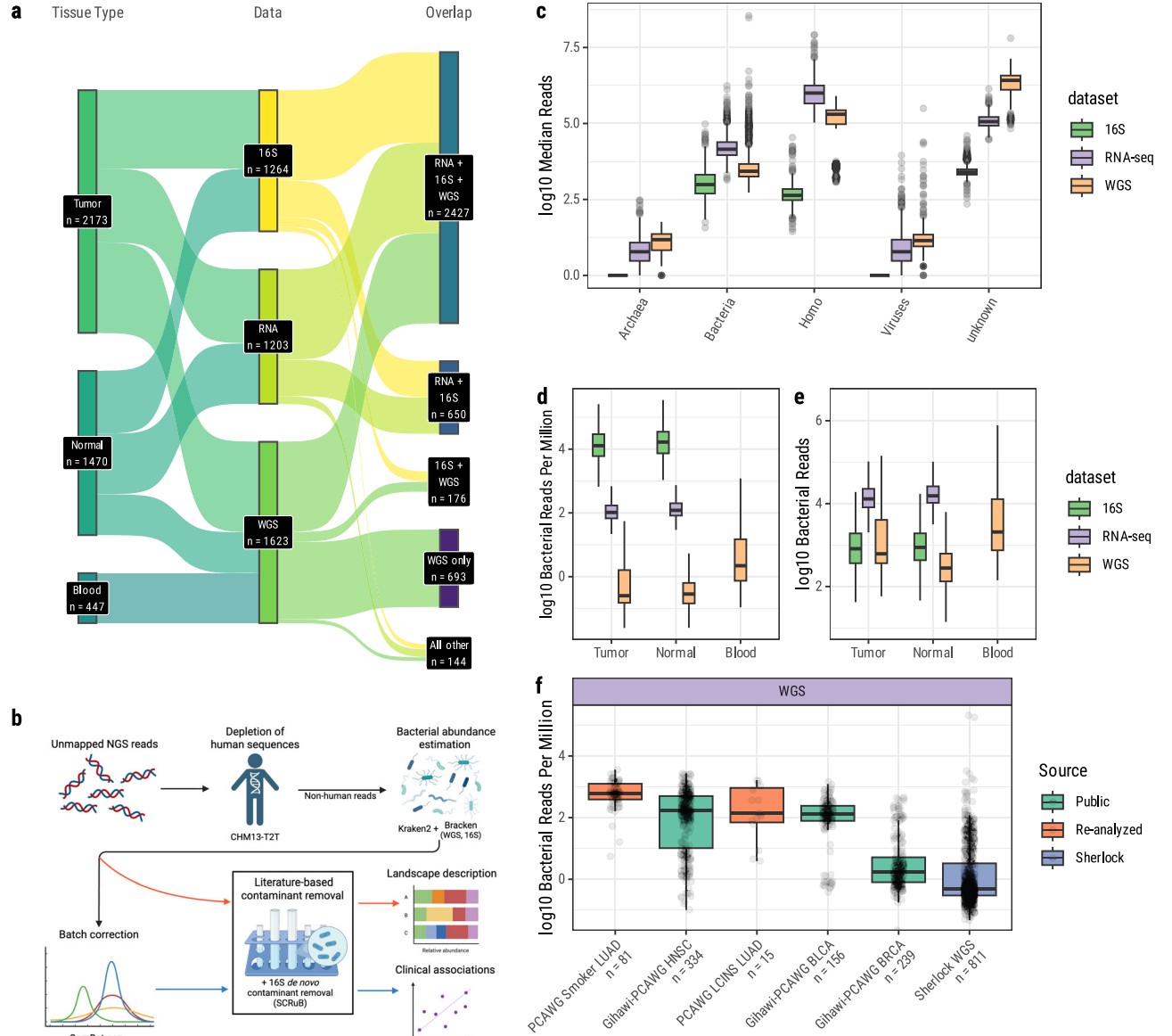

**Fig. 1 | General overview of the pipeline and dataset. a** Count of samples per combinations of sequencing platforms, by biospecimen type. **b** Overview of the analytical pipeline used for this study. Bracken abundance estimation was used only with WGS (combining this study and Zhang et al.) and 16S. After decontamination, read counts above the genus level were recursively adjusted ("Methods"). Created in BioRender. McElderry, J. (2025) https://BioRender.com/8kkrqgu. **c** Total reads assigned to different domains and to the human genome (WGS $n = 1176$; RNA-seq $n = 1203$, 16S $n = 1264$). **d** $\log_{10}$ bacterial reads per million, including human and other sequences, by sequencing modality and tissue type (WGS $n = 811$ tumors, 365 normal lung, 447 blood samples; RNA-seq $n = 661$ tumors, 542 normal

lung samples; 16S $n = 701$ tumors, 563 normal lung samples). **e** $\log_{10}$ absolute bacterial read counts by sequencing modality and tissue type (WGS $n = 811$ tumors, 365 normal lung, 447 blood samples; RNA-seq $n = 661$ tumors, 542 normal lung samples; 16S $n = 701$ tumors, 563 normal lung samples). **f** Comparison of $\log_{10}$ per-million genus-level bacterial reads in the WGS dataset compared to WGS from other studies. Boxplot centers, upper and lower bounds, and whiskers represent median, upper and lower quartiles, and quartiles ± 1.5 inter-quartile range, respectively. WGS whole genome sequencing, Rna-seq RNA sequencing, 16S 16S rRNA gene sequencing.

Despite rigorous filtering to remove human reads, many una-ligned reads in all datasets were assigned by Kraken2 to the human genome (median 48.1%, 6.4%, 8.3% in RNA-seq, WGS, 16S, respectively) (Fig. 1c and Supplementary Data 2–4). These reads likely originate from imperfect mapping of human[61], often repetitive, reads to the human genome. In 16S samples, many human reads originate from the mitochondrial genome, which contains a 16S rRNA gene that may be amplified off-target in 16S experiments[66]. RNA-seq samples contained the most human reads after alignment, perhaps in part due to the relative difficulty of filtering spliced human RNA sequences via mapping.

Many reads were of unknown origin (median 4.9%, 86.7%, 55.0% in RNA-seq, WGS, 16S, respectively), likely originating from sequencing artifacts, short sequences that could not be confidently assigned, or reads from microbes with incomplete reference genomes. Almost all taxonomically assigned, non-human reads were bacterial (median 99.9%, 98.7%, 100% among non-human reads in RNA, WGS, 16S, respectively), thus we focused our downstream analyses solely on the bacterial component of the datasets.

We next generated two datasets from the bacterial abundances: one with batch correction applied using ComBat-Seq, and one without batch correction. Reads without batch correction were used solely to

**Table 2 | Read depth statistics per sequencing modality and sample type**

| Tumor-normal status | n | mean | median | Standard deviation | Range Low | Range High |
|---|---|---|---|---|---|---|
| 16S | | | | | | |
| Tumor | 701 | 66,917 | 61,644 | 31,947 | 4024 | 199,104 |
| Normal | 563 | 57,628 | 58,308 | 20,127 | 5166 | 194,827 |
| RNAseq | | | | | | |
| Tumor | 661 | 129,635,580 | 127,478,501 | 38,900,833 | 22,429,409 | 610,965,375 |
| Normal | 542 | 130,148,355 | 126,346,728 | 37,999,097 | 20,866,863 | 471,422,274 |
| WGS | | | | | | |
| Blood | 447 | 942,996,840 | 908,564,834 | 189,070,286 | 680,579,372 | 1,814,647,436 |
| Tumor | 811 | 2,348,912,898 | 2,292,893,294 | 330,902,893 | 727,878,974 | 4,382,840,738 |
| Normal | 365 | 941,838,237 | 925,300,124 | 166,991,678 | 659,749,314 | 2,299,531,502 |

describe the landscape of the lung cancer microbiome as batch correction can, in some cases, greatly inflate the abundances of rare bacteria[61]. Batch corrected data were used for all statistical associations between the microbiome and clinical or demographic features. WGS associations were performed separately for samples sequenced for this study ($n = 1246$) and samples from our previous study ($n = 377$, Zhang et al.)[58] to account for a strong batch effect (Supplementary Fig. 1). Results from these two WGS data subsets were analyzed separately and combined as a meta-analysis downstream. For 16S data, abundances were not batch corrected as these samples did not show evidence of strong batch effects.

Both batch corrected and uncorrected abundances were then decontaminated in silico. For 16S samples, PCR negative controls were used to calculate bacterial contamination fractions with the SCRuB[67] algorithm, using PCR well location information to track well-to-well leakage. The WGS and RNA-seq datasets were originally collected for studies on human genomics/transcriptomics and therefore did not have paired negative controls, as this is not standard for non-metagenomic experiments. Despite this limitation, we sought to include both datasets as complementary data together with our 16S dataset to corroborate any findings. In all datasets, we performed literature-based decontamination by removing bacterial genera that are found to frequently contaminate NGS experiments[68] and have not been known to colonize human microbiomes[69] ("Methods"). Removal of reads at the genus level was recursively propagated[70] to higher taxonomic ranks to remove contamination at all levels of the taxonomy.

The raw composition of the microbiome at the phylum and genus levels is shown in Supplementary Fig. 2a. Prior to decontamination, we observed minimal correlation between sequencing platforms within the same samples (Supplementary Fig. 2c–f). Following batch correction and decontamination, phylum-level relative abundances and genus-level Shannon alpha diversity were significantly, but weakly, correlated across all datasets (alpha diversity Pearson $R$ values between 0.15–0.33, phylum-level abundances Pearson $R$ values between 0.0 and 0.27) (Supplementary Fig. 3). Furthermore, within-subject beta diversity accounted for a high percentage of overall variance among all samples (Permutational Multivariate ANOVA, 999 iterations, $p = 0.001$, $R^2 = 0.455$; Supplementary Data 5). This indicates that although microbiome composition and diversity results differ across sequencing modalities, the microbiome composition per subject, relative to other subjects, is similar across datasets.

**The lung cancer microbiome has low biomass across all data types**

16S samples had the most bacterial reads per million, as expected due to the targeted nature of 16S rRNA sequencing, followed by RNA-seq, and lastly WGS (Fig. 1d). After read filtering to remove contaminants, we observed low absolute bacterial read totals in WGS (median 344,

162, and 1440 bacterial reads in tumor, adjacent lung, and blood samples, respectively) and 16S sequencing samples (median 730 and 773 bacterial reads in tumor and adjacent normal tissue, respectively). The median numbers of bacterial reads in RNA-seq samples were 9080 and 11,053 in tumor and adjacent normal samples, respectively (Fig. 1e). Of note, WGS samples were sequenced to differing depth between tumor (median human genome coverage 87X) and normal lung tissue (median coverage 34×, read depth statistics for all samples provided in Table 2). We did not use microbiome data from WGS for alpha or beta diversity comparisons between tumor and normal lung tissue due to this difference, which could bias the results, and also due to the extremely low bacterial read depth in normal tissue.

To put these results into context, we compared the read counts of Sherlock WGS samples with those from the Pan-cancer analysis of whole genomes working group (PCAWG)[71] (Fig. 1f). We used the read counts from the PCAWG breast (BRCA), bladder (BLCA), and head and neck squamous cell carcinoma (HNSC) WGS samples re-analyzed by Gihawi et al.[61] and re-analyzed the PCAWG lung cancer WGS ($n = 96$, of which 81 from smokers, not reported in Gihawi et al.[61]). 16S samples were not included in this comparison as no public 16S data, both derived from lung tissue and including total read counts information, were available. We found that Sherlock WGS samples had lower genus-level bacterial reads in comparison to lung and other cancer types. Differences in DNA extraction and sequencing, as well as the different smoking status, may contribute to these findings, as PCAWG WGS is known to have batch-dependent bacterial contamination[72] (Fig. 1f).

For downstream statistical tests, RNA-seq samples with less than 500 reads were excluded to improve the reliability of associations. Due to the considerably lower read depth of 16S and WGS samples, this read cutoff was relaxed to 250 reads in 16S and 100 reads in WGS to preserve sample size. For intra-class correlation analyses (phylum-level relative abundances, alpha diversity, beta diversity), a cutoff of 250 reads was applied to all datasets to allow for valid comparisons.

**Microbiome composition across tissue and data types**

*Proteobacteria* (also known as *Pseudomonadota*, mean relative abundances per sequencing modality ranging 36.4–67.4%), *Actinobacteria* (also known as *Actinomycetota*, mean relative abundances 15.0–21.0%), and *Firmicutes* (also known as *Bacillota*, mean relative abundances 14.5–31.1%), were the most abundant phyla across all Sherlock datasets and biospecimen types (Fig. 2a). However, their mean relative abundances, particularly that of *Firmicutes*, varied substantially across sequencing modalities (Fig. 2b). Several bacterial genera were observed across all three datasets, e.g., *Acinetobacter* (mean relative abundance 5.9–8.6%), *Corynebacterium* (12.9–13.2%), *Pseudomonas* (2.7–23.9%), *Staphylococcus* (3.5–11.1%), and *Streptococcus* (2.3–3.9%; Fig. 2c and Supplementary Fig. 4). Notably, these were all among the top ten most abundant bacterial genera in a recent 16S sequencing

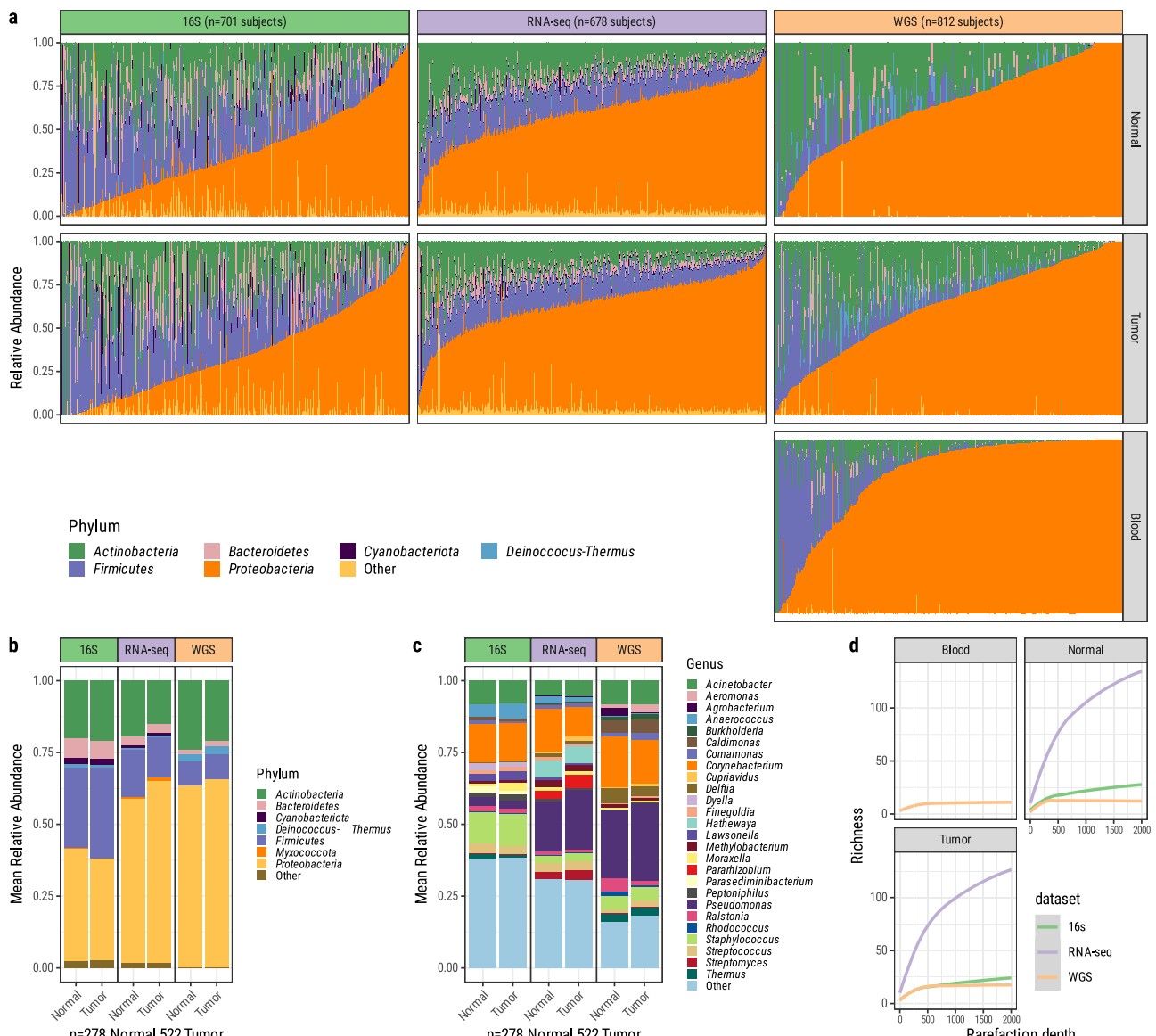

**Fig. 2 | Compositional overview of each dataset after decontamination.** **a** Overview of the phylum-level relative abundances for all samples in this dataset, ordered by abundance of Proteobacteria. **b** Mean phylum-level and **c** genus-level relative abundances by sequencing platform and tumor-normal status, including only samples that were sequenced across all three sequencing modalities. **d** Rarefaction curve showing the relationship between read depth and number of unique bacterial genera observed in 16S, RNA-seq, and WGS datasets across all tissue types. WGS whole genome sequencing, RNA-seq RNA sequencing, 16S 16S rRNA gene sequencing.

study of 245 lung tumors (43 never smokers)[35] with many negative controls and strict decontamination, thus demonstrating a degree of concordance between studies.

Among the three datasets, RNA-seq samples had the highest genus richness of all datasets, regardless of read sampling depth (Fig. 2d).

Comparing the tumor versus normal lung microbiome in 16S data, we identified few differentially abundant bacteria which were not significant after multiple testing correction (Fig. 3a; Supplementary Fig. 5a and Supplementary Data 6), and observed slightly decreased alpha diversity in tumor samples (mean diversity = 1.9) compared to paired normal tissues (mean diversity = 2.0, Wilcoxon $p = 0.0015$, Fig. 3b). Using RNA-seq, several bacterial genera were enriched in normal tissue compared with tumors (Fig. 3c; Supplementary Fig. 5b and Supplementary Data 7), and sample alpha diversity was marginally decreased in tumor tissue relative to paired normal tissues (Wilcoxon $p = 0.028$, mean diversity in tumors = 3.03,

normal tissue = 3.08) (Fig. 3d). Again using RNA-seq, we obtained similar results when we tested differential abundance of species within the most abundant genera (*Acinetobacter, Corynebacterium, Pseudomonas, Staphylococcus*, and *Streptococcus*): many *Corynebacterium* and *Staphylococcus* species were significantly enriched in normal lung tissue versus tumor tissue, and several species of *Pseudomonas* and *Acinetobacter* were marginally enriched in tumor tissue versus normal lung tissue when analyzed using ANCOM-BC (Supplementary Fig. 5c, d, and Supplementary Data 8).

We performed power calculations to derive the minimum effect sizes achieving 80% statistical power for detecting tumor/normal differences using either Bonferroni corrected $p$ value threshold or using 0.01 as $p$ value threshold ("Methods"). For tumor-normal comparisons, the minimum effect sizes are calculated as $\beta = 0.14$ for RNA-seq analysis ($\alpha = 3.9e-5$, Bonferroni correction, 1279 taxa) or $\beta = 0.096$ ($\alpha = 0.01$), and $\beta = 0.23$ for 16S sRNA analysis ($\alpha = 2.8e-4$, Bonferroni correction, 141 taxa) or $\beta = 0.17$

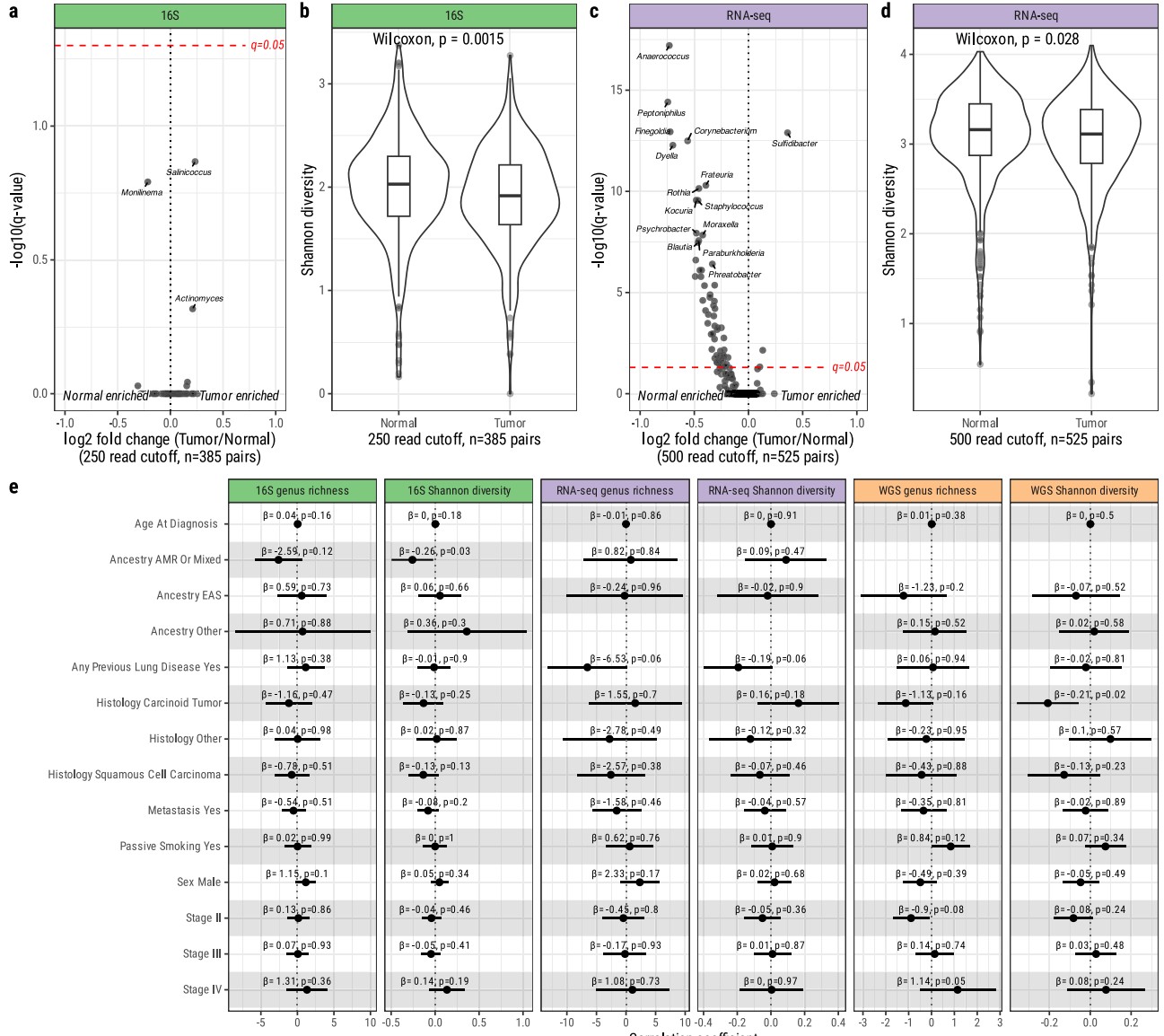

**Fig. 3 | Tumor-normal, clinical, and demographic associations with the microbiome.** **a** ANCOM-BC differential abundance results with the Holm method for multiple testing correction, and **b** comparison of Shannon alpha diversity between paired tumor and normal samples using genus-level 16S data ($n$ = 385 tumor-normal pairs) using a two-sided Wilcoxon test. Boxplot centers, upper and lower bounds, and whiskers represent median, upper and lower quartiles, and quartiles ± 1.5 inter-quartile range, respectively. **c** ANCOM-BC differential abundance results with Holm method multiple testing correction, and **d** comparison of Shannon alpha diversity between paired tumor and normal samples using genus-level RNA-seq data ($n$ = 525 tumor-normal pairs) using a two-sided Wilcoxon test. Boxplot centers, upper and lower bounds, and whiskers represent median, upper

and lower quartiles, and quartiles ± 1.5 inter-quartile range, respectively. **e** Genus-level alpha diversity and richness in tumors associated via generalized linear models with clinical features, adjusted for study site. RNA-seq ($n$ = 661), 16S ($n$ = 572), and meta-analyzed WGS ($n$ = 704) samples were rarefied to 500, 250, and 100 bacterial reads, respectively. Stage I tumors, adenocarcinoma histology, and European (EUR) ancestry serve as references. Unadjusted $p$ values are shown; all tests are non-significant (FDR > 0.05) after multiple testing correction. Points represent regression coefficient, error bars signify standard error. WGS whole genome sequencing, RNA-seq RNA sequencing, 16S 16S rRNA gene sequencing, AMR American, EAS East Asian.

($\alpha$ = 0.01). This suggests that we have sufficient statistical power to detect tumor-normal differences in the microbiome with modest effect sizes if they were present in our data.

We did not compare tumor WGS data with normal lung tissue WGS because of the different read depth between tumor and normal tissue, as previously stated.

## Microbiome characteristics in relation to demographic and clinical factors

We tested several factors in association with microbiome features. First, we examined the relationship between microbiome alpha

diversity versus clinical and demographic features. We observed variation in richness (Kruskal–Wallis, $p < 2.2e\text{-}16$) and diversity (Kruskal–Wallis, $p = 0.00033$) between study sites (Supplementary Fig. 6a, b). Other associations in all datasets were not significant after multiple testing corrections (Fig. 3e). Some associations were nominally significant. In WGS, stage IV tumors had increased genus richness relative to Stage I tumors (unadjusted $p = 0.05$, $\beta = 1.14$, 95% confidence interval = [−0.509, 2.79]), and carcinoid tumors had decreased alpha diversity compared to adenocarcinomas ($\beta = -0.21$, unadjusted $p = 0.02$, 95% confidence interval = [−0.36, −0.06]). In the 16S dataset, Native American/mixed ancestry patients had decreased tumor alpha

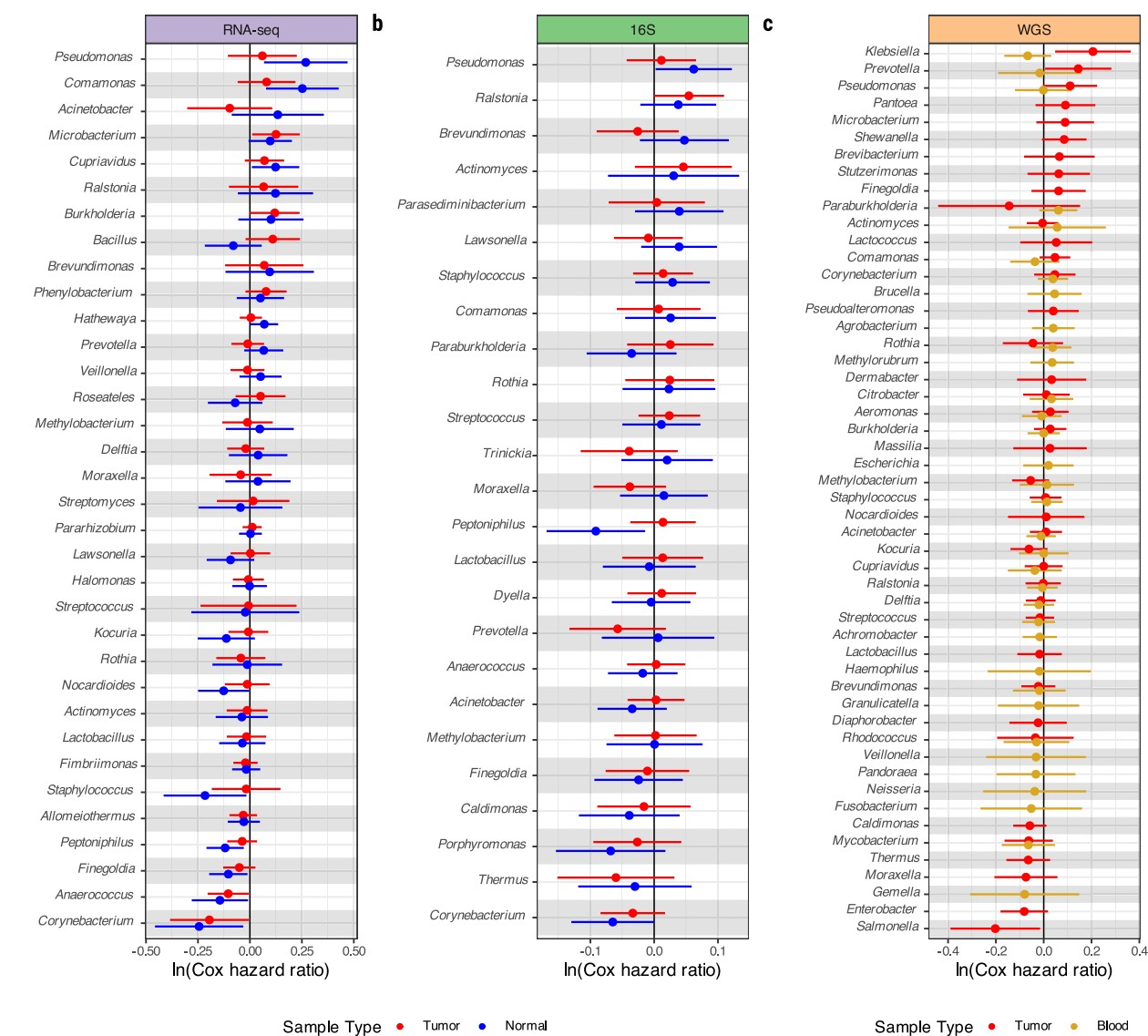

**Fig. 4 | Survival associations with individual bacterial taxa. a** Cox proportional hazard model of ten-year survival with RNA-seq bacterial relative abundances including genera with minimum 50 reads in 10% of samples, **b** 16S bacterial relative abundances including genera with minimum 10 reads in 10% of samples, and **c** meta-analyzed WGS bacterial relative abundances, including genera with minimum 10 reads in 10% of samples. All analyses were stratified by study site, age at diagnosis (age >65 or ≤65), and stage (stage I or stages II–IV), and further adjusted by histology and age in ten year categories. All associations are not significant (FDR > 0.05) after multiple testing correction. Points represent log Cox hazard ratio, error bars signify standard error. RNA-seq $n$ = 587 tumor samples, 482 normal samples; 16S $n$ = 488 tumor samples, 395 normal samples; WGS $n$ = 647 tumor samples, 375 blood samples. WGS whole genome sequencing, Rna-seq RNA sequencing, 16S 16S rRNA gene sequencing.

diversity relative to European patients (unadjusted $p$ = 0.03, β = −0.26, 95% confidence interval = [−0.50, −0.02]).

Measuring beta diversity, WGS and 16S datasets showed significant variation according to sample study site, while RNA-seq after batch correction was not significantly associated. The RNA-seq dataset showed small, significant differences according to tumor-normal status, age at diagnosis, histology, and vital status, but no differences were observed according to ancestry, sex, tumor stage, or development of metastases. No clinical and demographic variables were associated with beta-diversity using 16S or WGS data (Supplementary Fig. 6c).

Recent research has suggested that circulating bacterial DNA in blood may be associated with clinical outcomes, including in lung cancer[23–25]. To investigate this hypothesis, we tested associations between blood microbial diversity measures and lung cancer clinical features. We correlated relative abundances between paired tumor and blood samples at the phylum level to test the plausibility of

detecting lung bacteria in blood samples. Among the most prevalent phyla (*Proteobacteria*, *Actinobacteria*, *Firmicutes*, and *Bacteroidetes*), only abundance of phylum *Firmicutes* ($p$ < 2.2e-16, Pearson $R$ = 0.5) and *Proteobacteria* ($p$ = 9.1e-05, Pearson $R$ = 0.19) was significantly correlated between tumor and paired blood samples (Supplementary Fig. 7a). At the genus level, abundance of *Staphylococcus* (classified under phylum *Firmicutes*) was correlated between blood and tumor samples ($p$ < 2.2e-16, Pearson $R$ = 0.58). Genus richness and alpha diversity in blood samples were not associated with any tested clinical features, including lung cancer stage, histology, risk of recurrence, or vital status (Supplementary Fig. 7b), and beta diversity in blood was associated with sample study sites and weakly associated with vital status ($p$ = 0.026, $R^2$ = 0.008) and tumor stage ($p$ = 0.043, $R^2$ = 0.02) (Supplementary Fig. 7c).

Notably, using RNA-seq, 16S, and tumor and blood WGS data, we found no associations between genus-level relative abundances for

any bacteria, adjusted by histology and age in ten year categories, and overall survival, stratified by study site, age at diagnosis (age > 65, age ≤ 65), and tumor stage (Fig. 4). Similarly, no significant associations were observed for bacterial richness or alpha diversity with overall survival (Supplementary Fig. 8). Restricting survival analyses to lung adenocarcinomas only likewise produced no significant associations (Supplementary Fig. 9).

We performed power calculations to derive the minimum hazard ratios achieving 80% statistical power. Using the Bonferroni-corrected *p*-value thresholds (37 taxa for RNA-seq, 25 taxa for 16S rRNA, and 45 taxa for WGS), the minimum hazard ratio required to achieve 80% power was approximately 1.34 for all three platforms. When using a *p*-value threshold of 0.01, the minimum hazard ratios to achieve 80% power were approximately 1.27 for WGS, 1.30 for RNA-seq, and 1.29 for 16S rRNA. This indicates that if there were survival associations with modest effect sizes, we would have had sufficient statistical power to detect them.

We also tested whether RNA-seq and 16S bacterial richness or diversity was associated with immune cells by leveraging paired human transcriptomic data plus cell deconvolution methods. We noted a weak positive correlation between RNA-seq genus richness and log proportion of Th1 cells in tumor tissue, and a weak negative correlation between 16S Shannon diversity and log proportion of B-cells in normal tissues. Ultimately, however, we noticed no strong, consistent trends between datasets (Supplementary Fig. 10).

### The microbiome is not associated with human genomic features
We took advantage of the associated human whole-genome[58,59] data from these same samples and investigated whether major driver mutations or fusions, copy number alterations, kataegis, or mutational signatures in the human lung cancer genome were associated with microbiome richness and alpha diversity, adjusted for study site differences (Supplementary Data 9, and Supplementary Fig. 11a, b). All associations between the microbiome and genomic features were not significant after multiple testing correction (Supplementary Fig. 11c).

## Discussion
In the largest study of the LCINS microbiome to date using 16S sequencing, together with WGS and RNA-seq, we observed very low microbial abundance across over 4000 samples, and little evidence of association between the composition or diversity of the lung cancer microbiome and LCINS tumor characteristics, genomic features, and survival.

The bulk of research on the lung microbiome to date is derived from samples collected via BAL, and the consensus of these studies is that the healthy lung microbiome is composed mainly of oral and tracheal commensals (e.g., genera *Streptococcus* 15.7–38.7%, *Prevotella* 5–26.5%, *Veillonella* 3.8–4.0%, *Haemophilus* 0.02–15.5%, and *Neisseria* 6.5–9.3% among two BAL-based lung cancer studies[50,52], and among the highest abundance in several other studies in cancer[32,53] and non-cancer[28–30] patients). While these genera were present in our study, they summed to a small minority of the overall microbiome composition in all three data types (mean total relative abundance 4.0–5.8%). Instead, the highest abundance genera across all data types in tumors and normal lung were *Acinetobacter, Corynebacterium, Pseudomonas*, and *Staphylococcus*. These findings closely agree with a recent, highly decontaminated 16S sequencing dataset of 245 lung tumors (43 never smokers)[35] in which these genera were all among the top ten most abundant after decontamination. In blood WGS data, the most abundant bacteria were *Methylobacterium, Ralstonia, Burkholderia*, and *Pseudomonas*. Of note, the abundance of phyla *Firmicutes* and *Proteobacteria* and genus *Staphylococcus* was correlated between tumor samples and paired blood. These correlations may suggest migration of these bacteria from the lung to the blood, although translocation

from other organs and/or contaminations that could not be removed with the current approaches are always possible contributing factors. Nonetheless, we ultimately found no clinical associations with circulating bacteria.

RNA-seq data showed minor differences between tumor samples and paired adjacent normal tissue in alpha or beta diversity, and an enrichment of several human commensals in normal tissue (e.g., *Corynebacterium, Anaerococcus, Finegoldia*). Tumor tissues had slightly decreased alpha diversity compared to normal tissues in both the 16S and RNA-seq datasets. However, we noted no other robust associations of microbial abundances, richness, alpha diversity, or beta diversity with any available clinical features or patient survival, no associations between the microbiome and known tumor genomic features, and no consistent trends in microbiome-immune system crosstalk.

There are several limitations in this study. First, our normal lung tissue samples are only from lung cancer patients since lung tissue from healthy individuals can rarely be collected. Thus, we may have missed differences in the lung microbiome between healthy individuals and cancer patients. This study provides a snapshot of the microbiome at the time of tumor resection, and our samples were treatment-naïve, so we could not investigate the role of the microbiome on treatment response. This study lacks negative controls for RNA-seq and WGS datasets, which limits the identification of contaminating bacteria in these datasets. However, incomplete decontamination is more likely to result in false-positive than false-negative associations[61,73]. Furthermore, we leveraged state-of-the-art decontamination algorithms using negative controls in our 16S dataset and likewise produced no significant associations. Lastly, removing additional bacteria would unlikely result in positive associations given that we already have sufficient statistical power to detect associations with even modest effect sizes.

Every null result should be interpreted with caution. As methods for bacterial sequencing and microbiome analysis evolve to better accommodate low biomass samples, it is possible that a role for the lung microbiome in cancer could be found in the future. But, as it stands, after applying multi-omics datasets with rigorous quality control and state-of-the-art analytical methods in 4090 samples across 940 patients, the lung cancer microbiome does not appear to have a dominant role in LCINS.

## Methods
### Ethics declaration
The NCI exclusively received de-identified samples and data from collaborating centers, had no direct interaction with study subjects, and did not use or generate any identifiable private information, therefore, the Sherlock-*Lung* study was classified as "Not Human Subject Research (NHSR)" according to the Federal Common Rule (45 CFR 46; e). Some tissue specimens were obtained from the IUCPQ Tissue Bank, site of the Quebec Respiratory Health Network Biobank or the FQRS (www.tissuebank.ca) in compliance with Institutional Review Board-approved management modalities. Some samples and data from patients included in this study were provided by the INCLIVA Biobank (PT17/0015/0049), integrated in the Spanish National Biobanks Network and in the Valencian Biobanking Network, and they were processed following standard operating procedures with the appropriate approval of the Ethics and Scientific Committees. All collaborating centers obtained informed consent for publication of human data from participants under protocols approved by their respective Institutional Review Boards.

### Sample collection and handling
Samples were collected as described in previous Sherlock-*Lung* publications[57,58]. We collected tumor samples from 940 patients with histologically confirmed lung cancer from various geographical

regions: 220 from Taiwan; 208 from International Agency for Research on Cancer, Lyon, France, collected in Russia, Czech Republic, Romania, Serbia, and Poland; 133 from Hong Kong; 113 from Quebec City, Canada; 78 from Nice, France; 72 from Toronto, Canada; 26 from Massachusetts, USA; 22 from Connecticut, USA; 18 from Mexico City, Mexico; 13 from New York, USA; 13 from Minnesota, USA; 11 from Florida, USA; 9 from Valencia, Spain; and 4 from Lima, Peru. Fresh frozen tumor tissue and matched whole blood samples or fresh frozen normal lung tissue (collected at least 3 cm away from the tumor when possible) were obtained from these treatment-naïve patients. Genetic ancestry information was defined using WGS by clustering with the 1000 Genome Project (1KGR) reference panel with VerifyBamID2[74]. In the absence of WGS data, we relied on self-reported ancestry. For each patient, we reported the geographical location where the cancer was diagnosed.

We adhered to strict sample selection criteria:

1) Contamination and relatedness: cross-sample contamination was kept below 1% using Conpair[75], and relatedness was maintained below 0.2 using Somalier[76].
2) Copy number analysis: subjects with abnormal copy number profiles in normal samples were excluded, as determined by Battenberg[77].
3) Mutational signatures: tumor samples exhibiting mutational signatures SBS7 (associated with ultraviolet light exposure) or SBS31 (associated with platinum chemotherapy) were excluded.
4) WGS quality control: tumor samples with a total genomic alteration count of <100 or <1000 and an NRPCC (number of reads per clonal copy) <10 were excluded.

These stringent criteria were consistently applied to ensure data robustness and reliability in the Sherlock-*Lung* study.

## Whole-genome sequencing

WGS library construction was carried out as previously reported[58,59]. Briefly, frozen tumor tissue along with matched blood or normal tissue samples were immediately placed into 1 ml of 0.2 mg/ml Proteinase K (Qiagen) in DNA lysis buffer (10 mM Tris-Cl, pH 8.0; 0.1 M EDTA, pH 8.0; 0.5% SDS) and incubated for 24 h at 56 °C with shaking at 850 rpm in a Thermomixer R (Eppendorf) until completely lysed. Genomic DNA was extracted from fresh frozen tissue using the QIAmp DNA Mini Kit (Qiagen) following the manufacturer's protocol. Each sample was eluted in 200 µl AE buffer, and DNA concentration was measured using a Nanodrop spectrophotometer. All DNA samples were aliquoted and stored at −80 °C until needed.

DNA was quantified using the QuantiFluor® dsDNA System (Promega Corporation, USA). DNA standardized to a concentration of 25 ng/µl and underwent fragment analysis using the AmpFLSTR™ Identifiler™ PCR Amplification Kit (ThermoFisher Scientific, USA). DNA samples were required to meet minimum mass and concentration thresholds for each assay and show no evidence of contamination or profile discordance in the Identifiler assay. Samples that met these criteria were aliquoted at the appropriate mass needed for downstream assay processing.

The Broad Institute (https://www.broadinstitute.org) performed WGS on the Novaseq6000 platform using Illumina protocols for 2 × 150 bp paired-end sequencing in 1246 (this study) and the Illumina HiSeq X platform (*n* = 377) for our previous publication[58]. FASTQ files were generated post-Illumina base-calling. These paired FASTQ files were converted into unmapped BAM files using the GATK pipeline (https://github.com/gatk-workflows/seq-format-conversion) and were then processed using GATK on the cloud-based TERRA workspaces platform (https://app.terra.bio). The sequencing data were then aligned to the human reference genome GATK-GRCh38, and the resulting aligned BAM files were transferred to the NIH HPC system (https://hpc.nih.gov) for downstream analyses.

## RNA sequencing

RNA-seq was performed using the Illumina NovaSeq6000 platform and Roche KAPA RNA HyperPrep with RiboErase protocol, generating 2 × 151 bp paired-end reads. For human transcriptomics analyses, FASTQ files were aligned to the human reference genome GATK-GRCh38 using STAR[78] (v2.7.3), and were quantified using HTSeq(v2.0.4)[79] and GENCODE v35[80]. Counts data were batch corrected with ComBat-Seq[81], followed by TMM normalization using DESeq2[82].

## 16S microbiome sequencing

For each sample, 100 ng of DNA, utilizing Quant-iT PicoGreen dsDNA (Thermo Fisher Scientific, Waltham, MA) quantitation, is split into 50 ng (5 ng/µL) aliquots for two separate PCR reactions. PCR was performed in 25 µL reaction volumes consisting of: 50 ng (10 µL) of DNA, 10 µL of 2× PlatinumTM Hot Start PCR Master Mix (ThermoFisher Scientific), 3 µL of MBG Water, and 2 µL of the 5 µM 16S rRNA v4 (515f-806r) barcoded primer mix, comprised of equimolar forward and reverse primer pairs targeting the V4 region of the 16S rRNA gene[83]. Controls without input DNA were also included for PCR with the same reaction volumes, including a "water" control with 10 µL of MBG water in place of 10 µL of DNA, and a "no template" control with no DNA or added water. 515f forward PCR primer sequence was:

AATGATACGGCGACCACCGAGATCTACAC    TATGGTAATT    GT GTGCCAGCMGCCGCGGTAA

consisting of the 5′ Illumina adapter, forward primer pad, forward primer linker and forward primer. 806r reverse PCR primer sequence was:

CAAGCAGAAGACGGCATACGAGAT XXXXXXXXXXXX AGTCAGT-CAG CC GGACTACHVGGGTWTCTAAT

consisting of the reverse complement of the 3′ Illumina adapter, Golay barcode (12 bp barcode identifier generated specifically for this primer set to support multiplexing of samples), reverse primer pad, reverse primer linker and reverse primer (Integrated DNA Technologies, Coralville, IA). Thermal cycling was performed with the following PCR conditions: 94° C hold for 3 min, denature at 94° C for 45 s, anneal at 50° C for 1 min, extend at 72° C for 1 min 30 s for 25 cycles, followed by a 72° C hold for 10 min.

Sample PCR replicates were then pooled and purified using a 1:1 AMPure XP (Beckman Coulter Genomics, Danvers, MA) ratio, performing the final elution in 30 µL of Buffer EB (Qiagen, Germantown, MD). Amplified sample libraries were quantified using Quant-iT PicoGreen dsDNA Reagent (ThermoFisher Scientific, Waltham, MA) and up to 192, with unique barcoded adapters, were combined in equal amounts (100 ng each) and pools normalized to 10 nM with Buffer EB for pooled sequencing.

Sequencing was performed at the Cancer Genomics Research Laboratory using the Illumina MiSeq v2, 500 cycle kit (Illumina, San Diego, CA, USA) following the manufacturer's protocol[84] with the following modifications: pooled libraries were diluted to 5 pM in a serial dilution, and 25% denatured 5 pM PhiX was spiked-in and added to the "load sample" well. 3.4 µl of index sequencing primer at 100 mM, 3.4 µl of Read 1 Sequencing primer at 100 mM and 3.4 µl of Read 2 Sequencing Primer at 100 µM was added to wells 13, 12, and 14 of the MiSeq sequencing cartridge. 2 × 251 paired end sequencing was performed on the MiSeq, with up to 192 samples per run.

## Taxonomic classification of non-human reads

For classification of RNA-seq and WGS, unaligned read pairs were extracted from GATK-GRCh38-aligned bam files. To remove additional human reads, these reads were then realigned to the CHM13 T2T genome ref. 63, using bwa-mem[85] (v0.7.17) to align WGS and 16S reads, and hisat[86](v2.2.1-ngs3.0.1) to align RNA-seq reads. Unaligned read pairs were extracted from this alignment. Reads were then trimmed

using Trimmomatic[87] to remove trailing bases with average quality score less than 10 using a sliding window. Reads smaller than 45 bp after trimming were discarded.

Taxonomic assignment of reads was performed with Kraken2[64] (v2.1.2) using the Kraken2 standard database plus fungal and protozoan genomes downloaded on June 5th, 2023. For taxonomic assignment of RNA-seq reads, the human transcriptome was also included in the database to detect unaligned human reads spanning splice junctions. WGS read counts at the genus level were adjusted using Bracken[65], with a minimum of 2 reads per genus required prior to readjustment, and genera with single reads were discarded. Bacterial genera with fewer than 5 assigned reads in RNA-seq samples were discarded to remove false-positive assignments. Bracken was not used to adjust RNA-seq read counts, as we reasoned that Bracken's genome uniqueness statistic assumes roughly even genome coverage, which may be violated in cases where specific bacterial transcripts are highly upregulated.

Reads from 16S rRNA gene sequencing were taxonomically classified with Kraken2 using a Kraken2 database created from downloaded 16S gene sequences from NCBI plus the human genome GRCh38.p14. This database has the advantage of using identical taxonomies with the Kraken2 standard database, which facilitates comparison between sequencing platforms. 16S sequences were assigned with Kraken2 using a confidence threshold of 0.02 due to the high degree of similarity between 16S rRNA genes at the genus level. Genus-level read counts were then adjusted using Bracken[65] (v2.8) with a requirement of two reads per genus prior to readjustment.

### In silico sequencing and Kraken2 confidence threshold identification

Using InSilicoSeq[88] (v1.0), one million HiSeq reads were simulated from GATK-GRCh38 with uniform coverage. These reads were mapped back to GATK-GRCh38 using bwa-mem[85] (v0.7.17), then unaligned reads were extracted and pooled with 50,000 total reads simulated in the same manner from the genomes of eleven human-associated bacteria: *E. coli* (ASM584v2), *Pseudomonas aeruginosa* (ASM676v1), *Prevotella melaninogenica* (ASM14440v1), *Rothia mucilaginosa* (ASM17561v1), *Haemophilus parainfluenzae* (ASM19140v1), *Klebsiella pneumoniae* (ASM24018v2), *Staphylococcus epidermidis* (ASM609437v1), *Moraxella osloensis* (ASM155395v1), *Cutibacterium acnes* (ASM37670v1), *Streptococcus oralis* (46338_H01), and *Corynebacterium tuberculostearicum* (ASM1672836v1). These reads were taxonomically assigned using Kraken2 with default settings, and the percentage confidence with which each read was classified was calculated. A confidence threshold of 10% was chosen as, at this level, all simulated bacterial species were identifiable and most false-positive classifications would be excluded (Supplementary Fig. 12).

### Decontamination

16S sequencing samples were decontaminated on a plate-by-plate basis using the SCRuB algorithm[67] with default parameters and PCR well location information to track well leakage (Supplementary Fig. 13).

For WGS, previous studies have used paired blood samples from whole-genome sequencing experiments to flag contaminants under the assumption that tissue-associated microbes should be statistically more prevalent in tissue compared to paired blood[70]. Under this hypothesis, bacteria that are equally prevalent in blood and in paired tissue would be considered contamination. However, recent research on the human blood microbiome, with special attention paid to contaminant control, has indicated that the blood contains low levels of transient bacterial DNA, including from commensals previously associated with the oral/lung microbiomes[89]. Further, circulating microbial DNA, likely from tumor tissue, has been suggested as a biomarker for lung cancer detection[24]. Finally, low bacterial read depth in the WGS

samples greatly reduces the overall sensitivity of such a comparison. Thus, we chose not to use this method for decontamination in this study. Other decontamination methods, such as heuristic-based approaches[89] or the popular *decontam*[90] algorithm, were considered for the WGS and RNA-seq datasets. However, assumptions made by the frequency-based method of decontam are not valid in low-biomass environments[90].

In all datasets, bacterial genera identified as frequent NGS contaminants were removed using a list compiled by Salter et al.[68], and cross-referenced with a list of human-associated bacterial pathogens[69]. Bacterial genera identified as frequent NGS contaminators that encompassed two or more human-associated species were rescued to avoid discarding possibly true-positive reads. All other frequent NGS contaminators were discarded (Supplementary Data 10).

Genus *Cutibacterium* was also removed after reviewing the data. *Cutibacterium* is one of the most common skin commensals and a frequent contaminator of NGS experiments[68]. *Cutibacterium* was universally prevalent and highly abundant in our RNA-seq and WGS datasets, but infrequently observed in our 16S sequencing samples. Its removal from the WGS dataset considerably improved alpha diversity correlations and composition correlations between samples sequenced via both WGS and 16S sequencing. Furthermore, a recent lung cancer microbiome study with many negative controls showed minimal presence of *Cutibacterium* in lung tissue after decontamination[35]. For these reasons, *Cutibacterium* was identified as a likely contaminant and removed from the dataset. Several other skin-associated bacteria, such as *Corynebacterium* and *Staphylococcus*, did not share these properties and were therefore not removed from the dataset: they were prevalent across all datasets, they were observed in the same decontaminated lung microbiome dataset referenced above[35], and they are additionally associated with the nasal microbiome[91] and, more generally, humid microenvironments[92].

After decontamination at the genus level, reads were then adjusted at higher levels in the taxonomy as described by Dohlman et al.[70] Briefly, the number of reads assigned to a given bacterial OTU was multiplied by the percentage of non-contaminant reads at the next lowest taxonomic level within that OTU (e.g., family level reads are adjusted by multiplying the number of reads assigned to family X by the proportion of genus-level, non-contaminant reads within family X). This process was used recursively to decontaminate from the family level to the top of the taxonomy.

### Batch correction

WGS and RNA-seq genus- and phylum-level raw abundances (i.e., with no decontamination applied) were corrected to remove batch effects using Combat-Seq. Prior to batch correction, bacterial OTUs with prevalence less than 1% were removed in both datasets. For WGS, DNA extraction batch was used as the adjustment variable, and no biological variable was set, as DNA extraction batch was partially confounded with biosample type. All RNA for this study was extracted at the same laboratory, and was therefore corrected with study site as the batch variable and tumor-normal status as the biological variable. Following batch correction, decontamination of batch-corrected counts was applied as previously described at the genus level.

16S samples were left uncorrected as these samples did not show evidence of strong batch effects (Supplementary Fig. 1f).

### Differential abundance

Differential abundance was analyzed using the ALDEx2[93] and the ANCOM-BC[94] R packages. Bacterial taxa with a prevalence of less than 5% were discarded. For tumor versus normal differential abundance analysis, only subjects with paired tumor and normal lung tissue were included. For ANCOM-BC analysis, both study site and tumor-normal status were included in the differential abundance model to adjust for lingering batch effects.

## Microbiome diversity analyses

Microbiome diversity analyses were performed using the R package *vegan*. Genus richness was calculated as the expected number of unique bacterial genera at the specified rarefaction depths per sequencing modality. Alpha diversity analysis was performed using the Shannon index. Samples were randomly sampled to the appropriate rarefaction depth 100 times, and the median alpha diversity per sample over these 100 iterations was used downstream for alpha diversity calculations.

Beta diversity was calculated using Bray–Curtis distances with 50 random rarefaction sampling iterations at the previously specified sampling depths. Association of clinical variables with beta diversity was performed via permutational multivariate ANOVA analysis[95,96], implemented in the *adonis2* function, to find the marginal variance explained by each variable over 999 permutations.

## Survival analyses

For survival analyses, Cox proportional hazards models were fit with time since diagnosis as the time scale. Follow-up ended at death (overall survival), administrative censoring, or loss to follow-up. All survival times were censored at ten years for survival associations. The baseline hazards were stratified by study site, tumor stage, and age at diagnosis (age > 65, age ≤ 65). Tumor stages II, III, and IV were combined for more robust inference. Cox proportional hazards models were further adjusted for age at diagnosis in ten year categories.

For survival associations with individual bacterial abundances, only bacterial genera with at least 50 reads in RNA-seq in at least ten percent of samples were included (this read cutoff was relaxed to 10 in 16S and WGS to account for the lower read depth), and read counts were transformed using center log ratio transformation[97] with 0.05 added as pseudo-counts to the reads matrix.

## Statistical analyses

All statistical analyses were performed in R version 4.5.1. False discovery rates were calculated using the Benjamini-Hochberg method[98] for multiple hypothesis testing correction. For comparisons of continuous variables, Wilcoxon rank sums tests were employed unless otherwise noted. Prior to correlating individual bacterial relative abundances, read counts were transformed per-sample using center log ratio transformation with 0.05 pseudo-counts added.

Meta-analysis of generalized linear models and Cox models for WGS was performed using a fixed effect model, and $p$ values were calculated using Fisher's combined probability test. Meta-analysis of beta-diversity was performed by averaging $R^2$ between the two data subsets, and $p$ values were calculated using Fisher's combined probability test.

## Power calculations

***Power for detecting difference between tumor and matched normal tissue samples.*** For the $i^{th}$ subject, let $x_i^+$ be the measurement of the tumor sample and $x_i^-$ be the measurement of the adjacent normal tissue sample. We tested the null hypothesis that the measurement does not differ between tumor and normal tissue samples using a paired $t$-test. For the $i^{th}$ subject, define difference $\delta_i = x_i^+ - x_i^-$ and let $\hat{\sigma}^2 = \mathrm{var}(\delta_i)$. Let effect size $\beta = E(\delta)/\sigma$, the expected difference between tumor and normal samples, normalized by standard deviation. The noncentrality parameter of the paired t-test $z$ is approximately $\xi = \beta\sqrt{n}$, where $n$ is the number of subjects. Since sample size $n$ is reasonably large, we use normal distribution to approximate the power:

$$P(|z| > C_\alpha) = P(z > C_\alpha) + P(z < -C_\alpha)$$
$$= P(z - \xi > C_\alpha - \xi) + P(z - \xi < -C_\alpha - \xi) \quad (1)$$

which is simplified as $\Phi(\xi - C_\alpha) + \Phi(-\xi - C_\alpha)$. Here, $C_\alpha$ is the quantile corresponding to level $\alpha$, with $\alpha$ chosen by the Bonferroni correction or $\alpha = 0.01$. *Survival power analysis* For each of the platforms (WGS, RNA-seq, 16S rRNA), we conducted power simulations by conditioning on the observed distribution of survival times and the fraction of censoring, assuming non-informative censoring. We assumed that survival times followed a log-normal distribution, i.e., $\log S \sim N(\mu, \sigma^2)$, and estimated the parameters $\mu$ and $\sigma^2$ by maximizing the log-likelihood function using all available subjects with survival data. We then simulated a microbiome feature variable $x_i \sim N(0, 1)$ and generated event times from the proportional hazards model

$$H(S_i|z_i) = H_0(S_i)exp(\beta z_i) \quad (2)$$

using the inverse probability method described by Bender et al.[99]. A random fraction of subjects was selected for censoring to match the censoring rate observed in the real dataset. We applied Cox proportional hazards regression to derive the Wald statistic for testing the null hypothesis $H_0 : \beta = 0$. This simulation process was repeated 1000 times, and power was calculated as the proportion of simulations yielding a $p$ value below a specified threshold (either Bonferroni-corrected or $p = 0.01$).

## Identification of tumor somatic alterations

Somatic mutation calling was conducted using our previously established bioinformatics pipeline[58]. We utilized four distinct mutation calling algorithms for tumor-normal/blood paired analysis, including Strelka[100] (v.2.9.10), MuTect[101], MuTect2, and TNscope[102] within Sentieon's genomics software(v.202010.01). An ensemble method was employed to integrate the results from these different callers, followed by additional filtering to minimize false positives. Final mutation calls for both single nucleotide variants and indels had to meet the following criteria:

1) Read depth >12 in tumor samples and >6 in normal samples
2) Variant allele frequency <0.02 in normal samples
3) Overall allele frequency (AF) < 0.001 in multiple genetic databases, including 1000 Genomes (phase 3v.5), gnomAD exomes (v.2.1.1), and gnomAD genomes (v.3.0)[103]

For indel calling, only variants identified by at least three algorithms were retained. The IntOGen pipeline (v.2020.02.0123)[104], which integrates seven advanced computational methods, was used with default parameters to detect positive selection signals in the mutational patterns of driver genes across the cohort.

The Battenberg algorithm[77] (v.2.2.9) was applied to analyze somatic copy number alterations (SCNA). Initial SCNA profiles were generated, followed by an evaluation of the clonality of each segment, purity, and ploidy. SCNA profiles deemed low-quality after manual inspection underwent a refitting process, requiring new tumor purity and ploidy inputs, either estimated by ccube[105] (v.1.0) or recalculated from local copy number status. The Battenberg refitting procedures were iteratively executed until the final SCNA profile met manual validation criteria. GISTIC[106] (v.2.0) was used to identify recurrent copy number alterations at the gene level based on the major clonal copy number for each segmentation. Structural variants (SVs) were identified using Meerkat[107] (v.0.189) and Manta[108] (v.1.6.0), applying recommended filtering, and the union of these two callers was combined to create the final SV dataset.

## Tumor genomic driver mutation analysis

To identify driver mutations among the set of recognized driver genes, we applied a comprehensive and robust strategy, incorporating several key criteria: (a) the presence of truncating mutations specifically in genes classified as tumor suppressors, (b) the recurrence of missense mutations in at least 3 samples, (c) mutations labeled as "Likely drivers"

with a boostDM[109] score exceeding 0.5, (d) mutations classified as "Oncogenic" or "Likely Oncogenic" according to OncoKB[110], (e) mutations previously identified as drivers using TCGA MC3[111], and (f) missense mutations deemed "likely pathogenic" in genes annotated as tumor suppressors, as outlined by Cheng et al.[112]. Mutations meeting any one of these criteria were considered potential driver mutations.

## Mutational signature analysis

The methods for mutational signature analysis are as previously described[59]. Briefly, SigProfilerMatrixGenerator[113] was utilized to generate mutational matrices for all types of somatic mutations, including single base substitutions (SBS), doublet base substitutions (DBS), and indels (ID). De novo SBS, DBS, and ID signatures were extracted using SigProfilerExtractor[114] (v1.1.21) with default settings, normalizing to 10,000 mutations, and using the SBS-288, DBS-78, and ID-83 mutational contexts. Subsequently, de novo extracted signatures were decomposed into COSMICv3.4[115] reference signatures based on the GRCh38 reference genome. These decomposed signatures were assigned to individual samples using SigProfilerAssignment[116] (v0.1.1).

## RNA-seq cell-type deconvolution analysis

For evaluation of the immune component of each sample, we used a list of immune-cell marker genes that were previously benchmarked and found to perform optimally for immune cell deconvolution in non-small cell lung cancer[117,118]. Samples were scored for each cell type using the median logCPM expression value among all genes within each set of cell-specific markers.

## Reporting summary

Further information on research design is available in the Nature Portfolio Reporting Summary linked to this article.

## Data availability

Whole genome sequencing data used in this study is deposited in the dbGaP database under accession code phs001697.v2.p1[https://www.ncbi.nlm.nih.gov/projects/gap/cgi-bin/study.cgi?study_id=phs001697.v2.p1]. RNA-seq data used in this study is deposited in the dbGaP database under accession code phs003955.v1.p1. 16S rRNA gene sequencing data are deposited in the SRA database under BioProject accession code PRJNA1337178.

## Code availability

The bioinformatics pipeline can be accessed at https://github.com/jpmcelderry/Sherlock-microbiome.

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

## Acknowledgements

This work was supported by the Intramural Research Program of the National Cancer Institute, US National Institutes of Health (NIH) (project ZIACP101231 to M.T.L.). The contributions of the NIH authors were made as part of their official duties as NIH federal employees, are in compliance with agency policy requirements, and are considered Works of the United States Government. However, the findings and conclusions presented in this paper are those of the authors and do not necessarily reflect the views of the NIH or the U.S. Department of Health and Human Services. Additional funds were NIH grants R01ES032547-01, R01CA269919-01, and 1U01CA290479-01 to L.B.A., as well as by L.B.A.'s Packard Fellowship for Science and Engineering, and by NIH grant U01CA209414 to DCC to support the Boston Lung Cohort Study. Where authors are identified as personnel of the International Agency for Research on Cancer/World Health Organization, the authors alone are responsible for the views expressed in this article, and they do not necessarily represent the decisions, policy or views of the International Agency for Research on Cancer/World Health Organization. We want to particularly acknowledge the patients and the INCLIVA Biobank (PT17/0015/0049) integrated in the Spanish National Biobanks Network and in the Valencian Biobanking Network for their collaboration. This study was supported by the Health and Medical Research Fund of Hong Kong S.A.R., HMRF 03142856. The related studies of Taiwan site were supported by grants from the Ministry of Health and Welfare, Taiwan DOH97-TD-G-111-026 (C.A.H.), DOH98-TD-G-111-015 (C.A.H.), DOH99-TD-G-111-028 (C.A.H.); DOH97-TD-G-111-029 (C.Y.C.), DOH98-TD-G-111-018 (C.Y.C.), DOH99-TD-G-111-015 (C.Y.C.) and the Ministry of Science and Technology, Taiwan MOST109-2740-B-400-002 (C.A.H.), MOST110-2740-B-400-002 (C.A.H.), MOST111-2740-B-400-002 (C.A.H.). This work has been supported in part by the Tissue Core at the H. Lee Moffitt Cancer Center & Research Institute, a comprehensive cancer center designated by the National Cancer Institute and funded in part by a Moffitt Cancer Center Support Grant (no. P30-CA076292). The authors would like to thank the team at the IUCPQ site of the Quebec Respiratory Health Network Biobank of the FRQS for their valuable assistance, and would like to thank the staff at Harvard University, Yale University, Roswell Park Cancer Institute and Roswell PI, Instituto Nacional de Cancerologia, Nice University Hospital Centre (Nice UHC)—University Côte d'Azur and the Nice Biobank CRB, Toronto University Health Network, and Mayo Clinic for their assistance providing samples and corresponding clinical data. The computational analyses reported in this manuscript have utilized the NIH high-performance Biowulf Cluster. We thank the study participants and the staff at Westat Inc. for their assistance in collecting samples and corresponding clinical data. We would like to thank Ruth Pfeiffer for her advice on survival analyses.

## Author contributions

Conceptualization: M.T.L. and J.P.M.; Methodology: J.P.M., J.Sh., T.Z., J.S., A.K., M.S., O.L., S.S., K.M.J., M.A.N., and M.T.L.; Formal analysis: J.P.M., T.Z., W.Z., E.V., C.C.A., B.Z., M.D-.G., D.C.W., L.B.A., J.Sh., and S.A-.S.; Pathology work: R.H., S-.R.Y., L.M.S., C.L., M.K.B., P.J., and W.D.T.; Management: P.Hoa.; Resources: L.M., O.G.A.R., E.S.E., J.M.S., M.B.S., S.S.Y., M.Ma., J.L., B.S., A.M., O.S., D.Z., I.H., V.J., D.M., S.M., M.S., M.K., Y.B., B.E.G.R., D.C.C., V.G., P.B., G.L., P.Hof., M.P.W., K.C.L., C-.Y.C., C.A.H., N.R., Q.L., M.T.L., and S.J.C.; Data curation: P.Hoa., T.Z., W.Z., C.H., F.J.C-.M., and M.Mi.; Writing, Original draft: J.P.M., J.Sh., and M.T.L.; Writing, Review & Editing: All authors; Visualization: J.P.M., T.Z., and M.T.L.; Supervision: M.T.L.

## Funding

## Competing interests

L.B.A. is a co-founder, C.S.O., scientific advisory member, and consultant for io9, has equity and receives income. The terms of this arrangement have been reviewed and approved by the University of California, San Diego in accordance with its conflict of interest policies. L.B.A. is also a compensated member of the scientific advisory board of Inocras. L.B.A.'s spouse is an employee of Biotheranostics. E.N.B. and L.B.A. declare U.S. provisional patent application filed with UCSD with serial numbers 63/269,033. L.B.A. also declares U.S. provisional applications filed with UCSD with serial numbers: 63/366,392; 63/289,601; 63/483,237; 63/412,835; and 63/492,348. L.B.A. is also an inventor of a US Patent 10,776,718 for source identification by non-negative matrix factorization. S.R.Y. has received consulting fees from AstraZeneca, Sanofi, Amgen, AbbVie, and Sanofi; received speaking fees from AstraZeneca, Medscape, PRIME Education, and Medical Learning Institute. All other authors declare that they have no competing interests.

## Additional information

John P. McElderry[1], Tongwu Zhang [1], Wei Zhao [1], Phuc H. Hoang [1], Samuel Anyaso-Samuel[1], Jian Sang[1], Azhar Khandekar[1,2], Caleb Hartman [1], Frank J. Colón-Matos[1], Mona Miraftab[1], Monjoy Saha [1], Olivia Lee [1], Sunandini Sharma[1], Kristine M. Jones [1,3], Bin Zhu [1], Marcos Díaz-Gay [2], Luis Mas[4], Oscar Arrieta[5], Eric S. Edell[6],

Jacobo Martínez Santamaría[7], Matthew B. Schabath [8], Sai Yendamuri[9], Marta Manczuk [10], Jolanta Lissowska [10], Beata Świątkowska[11], Anush Mukeria[12], Oxana Shangina[12], David Zaridze[12], Ivana Holcatova [13,14], Vladimir Janout[15], Dana Mates[16], Simona Ognjanovic[17], Milan Savic[18], Milica Kontic[19], Yohan Bossé [20], Bonnie E. Gould Rothberg[21], David C. Christiani [22,23], Valerie Gaborieau[24], Paul Brennan[24], Geoffrey Liu[25], Paul Hofman [26], Maria Pik Wong[27], Kin Chung Leung[28], Chih-Yi Chen[29,30], Chao Agnes Hsiung[31], Nathaniel Rothman[1], Charles Leduc[32], Marina K. Baine[33], William D. Travis[33], Lynette M. Sholl [34], Philippe Joubert[20], Robert Homer [35], Soo-Ryum Yang[33], Qing Lan [1], Martin A. Nowak[36,37], David C. Wedge [38], Ludmil B. Alexandrov [2], Stephen J. Chanock [1], Emily Vogtmann [1], Christian C. Abnet [1], Jianxin Shi [1] & Maria Teresa Landi[1] ✉

[1]Division of Cancer Epidemiology and Genetics, National Cancer Institute, Bethesda, MD, USA. [2]Department of Cellular and Molecular Medicine and Department of Bioengineering and Moores Cancer Center, UC San Diego, La Jolla, CA, USA. [3]Cancer Genomics Research Laboratory, Leidos Biomedical Research, Frederick National Laboratory for Cancer Research, Frederick, MD, USA. [4]Medical Oncology Unit, Instituto Nacional de Enfermedades Neoplásicas (INEN), Lima, Peru. [5]Thoracic Oncology Unit, Instituto Nacional de Cancerología (INCan), Mexico City, México. [6]Division of Pulmonary and Critical Care Medicine, Mayo Clinic, Rochester, MN, USA. [7]Biobanco IBSP-CV FISABIO; Red Valenciana de Biobancos, Cataluña, Valencia, Spain. [8]Department of Cancer Epidemiology, H. Lee Moffitt Cancer Center and Research Institute, Tampa, FL, USA. [9]Thoracic Surgery, Roswell Park Comprehensive Cancer Center, Buffalo, NY, USA. [10]Department of Cancer Epidemiology and Primary Prevention, Maria Skłodowska-Curie National Research Institute of Oncology, Warsaw, Poland. [11]Department of Environmental Epidemiology, Nofer Institute of Occupational Medicine, Łódź, Poland. [12]Department of Clinical Epidemiology, N.N. Blokhin National Medical Research Centre of Oncology, Moscow, Russia. [13]Institute of Public Health & Preventive Medicine, 2nd Faculty of Medicine, Charles University, Prague, Czechia. [14]Department of Oncology, 2nd Faculty of Medicine, Charles University and Motol University Hospital, Prague, Czechia. [15]Faculty of Health Sciences, Palacky University, Olomouc, Czechia. [16]Department of Occupational Health and Toxicology, National Center for Environmental Risk Monitoring, National Institute of Public Health, Bucharest, Romania. [17]International Organization for Cancer Prevention and Research (IOCPR), Belgrade, Serbia. [18]Department of Thoracic Surgery, Clinical Center of Serbia, Belgrade, Serbia. [19]Clinic of Pulmonology, Clinical Center of Serbia, Belgrade, Serbia. [20]Institut universitaire de cardiologie et de pneumologie de Québec, Université Laval, Québec City, QC, Canada. [21]UMMG Dept of Medicine - Hospital Medicine, Miller School of Medicine, Medical Campus, University of Miami, Miami, FL, USA. [22]Department of Environmental Health, Harvard T.H. Chan School of Public Health, Boston, MA, USA. [23]Department of Medicine, Massachusetts General Hospital, Boston, MA, USA. [24]Genomic Epidemiology Branch, International Agency for Research on Cancer (IARC/WHO), Lyon, France. [25]Division of Medical Oncology, Medicine, Princess Margaret Cancer Centre, Temerty Faculty of Medicine, Dalla Lana School of Public Health, University of Toronto, Toronto, ON, Canada. [26]Nice University Hospital Centre (Nice UHC) - University Côte d'Azur and the Nice Biobank CRB, Nice, France. [27]Queen Mary Hospital, The University of Hong Kong, Hong Kong, China. [28]Department of Pathology, The University of Hong Kong, Hong Kong, China. [29]Institute of Medicine, Chung Shan Medical University, Taichung, Taiwan. [30]Division of Thoracic Surgery, Department of Surgery, Chung Shan Medical University Hospital, Taichung, Taiwan. [31]Institute of Population Health Sciences, National Health Research Institutes, Zhunan, Taiwan. [32]Department of Pathology, Centre Hospitalier de l'Université de Montréal, Montreal, QC, Canada. [33]Department of Pathology and Laboratory Medicine, Memorial Sloan Kettering Cancer Center, New York, NY, USA. [34]Department of Pathology, Brigham and Women's Hospital, Boston, MA, USA. [35]Department of Pathology, Yale School of Medicine, New Haven, CT, USA. [36]Department of Mathematics, Harvard University, Cambridge, MA, USA. [37]Department of Organismic and Evolutionary Biology, Harvard University, Cambridge, MA, USA. [38]Manchester Cancer Research Centre, Division of Cancer Sciences, University of Manchester, Manchester, UK. ✉e-mail: landim@nih.gov

