## [Transparent Peer Review File · Nature Communications]

Microbiome analysis of 940 lung cancers in never-smokers reveals lack of clinically relevant associations

Corresponding Author: Dr Maria Teresa Landi

Version 0:

Reviewer comments:

Reviewer #1

(Remarks to the Author)

This manuscript represents an impressive and comprehensive investigation into the lung tissue microbiome in never-smoker lung cancer patients. The scale of the study, with nearly a thousand patients and multiple sequencing modalities, as well as the rigor of the contamination and batch effect controls, sets a new benchmark for the field. The negative results are particularly compelling given the thoroughness of the analysis and the authors' transparent reporting of their methods and findings.

I have only a few minor questions and suggestions that I believe could further strengthen the manuscript.

First, while the study includes both tissue and blood WGS data, I did not see an explicit analysis of the correlation between the microbial signals detected in blood and those found in lung tissue. The assumption that the circulating bacterial DNA detected in blood WGS originates from the lung tumor is a strong one, especially in the absence of direct evidence for lung barrier disruption in this patient population. It would be valuable to include an analysis or discussion of whether there is any meaningful correlation between the microbial profiles in blood and tissue. In the absence of such correlation, it is plausible that the microbial DNA detected in blood may originate from other sites, such as the gut or skin, rather than the lung itself.

Second, I am interested in whether the authors were able to assess the anatomical location of the tissue samples, specifically whether they could distinguish between samples from more proximal tissues versus those from the distal, alveolated parenchyma. Prior studies have shown that there is a gradient in bacterial biomass and community composition from the upper airways to the distal lung regions, with higher microbial biomass in more proximal sites, e.g. as shown in this study PMID: 33268457

If histological or anatomical data are available, it would be informative to know whether any such gradient was observed in this cohort, or if the bacterial signal was uniformly low across all sampled regions. Even a brief mention of this consideration, or a statement about the availability of such data, would add valuable context to the findings.

Finally, a landmark methodological paper on use of different biospecimens for lung microbiota analysis warrants citations in my view.

PMID: 31661299

(Remarks on code availability)

Reviewer #2

(Remarks to the Author)

In this impressive study, the authors seek to identify a lung cancer-associated microbiome signature by studying lung biopsy and blood samples from 940 patients. They use a combination of 16S, WGS and RNAseq to characterize the microbial communities of tumors and adjacent non-tumor and blood samples. Ultimately the study findings were negative- across platforms, alpha/beta diversity and individual genera did not associate with age, sex, ancestry, passive smoking, histology, stage, or survival. The study was sufficiently powered and the conclusions are believable. Generally, the methods are

sound, and the authors make a reasonable attempt to address batch effects and background contamination, although the lack of negative controls for WGS and RNAseq data is suboptimal.

All said, this is an important contribution to the literature and in this case, a negative result is valuable knowledge for the scientific community.

I have some suggestions for further improving this study:

- 1.The authors used the simple but somewhat outdated ALDEx2 algorithm for differential abundance analyses. Can the authors confirm that using a more modern and widely accepted tool such as ANCOM-BC (<https://www.nature.com/articles/s41467-020-17041-7>) does not globally change the primary study conclusions?
- 2.Can the authors please justify their minimum read depth thresholds?
- 3.Can the authors report effect sizes (e.g., log2FC and 95% Cis for enriched taxa)?
- 4.Can the authors please include read depth statistics (e.g., mean, range, std deviation of reads per sample)?
- 5.The lack of negative controls and background correction for the WGS and RNAseq datasets is somewhat concerning. The reason for this is understandable however given that the data analyzed were not initially collected for metagenomic analysis. The authors reasonably compensate for this by subtracting reads from a previously curated list of established metagenomic contaminants. The authors state however that they “manually” removed taxa. Can the authors clarify whether they used a systematic computational approach to remove the same blacklisted taxa (Supp Table 7?) from every sample, or whether they indeed manually subtracted taxa on a sample-by-sample basis?
- 6.The authors have provided only the submission reference numbers and not the accession numbers for their data, which are not accessible for review

(Remarks on code availability)

I did not run the code because the intermediate files necessary to do so are unavailable on the GitHub repository. Upon inspection, the code appears clean and well annotated.

It would be helpful if the authors provided taxonomic count data on their GitHub repo or in supplemental data tables so the R code in the repo could be run to reproduce the results/figures without de-nova re-alignment of the raw sequencing reads.

Version 1:

Reviewer comments:

Reviewer #1

(Remarks to the Author)

The authors have responded nicely to my comments and in my view, to the comments of the other reviewer. I maintain my high level of appreciation for this in-depth work with overall neutral findings for the role of lung microbiome in lung cancer of never smokers.

The only minor comment I have pertains to the lung-blood correlation analysis, which was done at the Phylum level. I am afraid that this can create some spurious association and lead the reader to conclude that the significant but weak correlation for Firmicutes taxa may suggest lung origin of the detected taxa in the blood. There are stronger competing explanations, such as subclinical gut translocation or skin contamination/translocation of Firmicutes; unless the authors pursue a comparison at a lower taxonomic level (e.g. genera) and show some matching between lung and blood taxa, I recommend staying away from oversimplification for a plausible lung-origin of detectable DNA in the circulation, or at a minimum acknowledging the competing explanations in the Discussion.

(Remarks on code availability)

Reviewer #2

(Remarks to the Author)

I thank the authors for thoroughly addressing my comments and requests. I believe this is a valuable study and is suitable for publication.

(Remarks on code availability)

Reviewer #1 (Remarks to the Author):

This manuscript represents an impressive and comprehensive investigation into the lung tissue microbiome in never-smoker lung cancer patients. The scale of the study, with nearly a thousand patients and multiple sequencing modalities, as well as the rigor of the contamination and batch effect controls, sets a new benchmark for the field. The negative results are particularly compelling given the thoroughness of the analysis and the authors' transparent reporting of their methods and findings.

We truly appreciate the reviewer's positive assessment of our work.

I have only a few minor questions and suggestions that I believe could further strengthen the manuscript.

First, while the study includes both tissue and blood WGS data, I did not see an explicit analysis of the correlation between the microbial signals detected in blood and those found in lung tissue. The assumption that the circulating bacterial DNA detected in blood WGS originates from the lung tumor is a strong one, especially in the absence of direct evidence for lung barrier disruption in this patient population. It would be valuable to include an analysis or discussion of whether there is any meaningful correlation between the microbial profiles in blood and tissue. In the absence of such correlation, it is plausible that the microbial DNA detected in blood may originate from other sites, such as the gut or skin, rather than the lung itself.

We thank the reviewer for the suggestion. We performed a correlation of the centered log-ratio abundances between paired tumor-blood samples at the phylum level and included the results as **Supplementary Figure 6a**. Phylum *Firmicutes* was significantly correlated between tumor and blood samples ($R=0.3$, $P=1.4e-06$). This suggests that this phylum in the blood could have originated from the lung tissue. We have added this finding in the manuscript's Results, Supplementary Figure 6a, and Discussion:

Results

“We correlated relative abundances between paired tumor and blood samples at the phylum level to test the plausibility of detecting lung bacteria in blood samples. Among the most prevalent phyla (*Proteobacteria*, *Actinobacteria*, *Firmicutes*, and *Bacteroidetes*), only abundance of phylum *Firmicutes* was significantly correlated between tumor and paired blood samples (Pearson $R=0.3$, $p=1.4e-06$)(**Supplementary Figure 6a**), suggesting that bacteria within this phylum may migrate from lung tissue into the blood.”

Discussion

“Of note, abundance of phylum Firmicutes was correlated between tumor samples and paired blood, which suggests that these bacteria may migrate from lung tissue to blood. Though this does support the plausibility of detecting lung microbiome signatures in the blood, we ultimately found no clinical associations with circulating bacteria.”

Second, I am interested in whether the authors were able to assess the anatomical location of the tissue samples, specifically whether they could distinguish between samples from more proximal tissues versus those from the distal, alveolated parenchyma. Prior studies have shown that there is a gradient in bacterial biomass and community composition from the upper airways to the distal lung regions, with higher microbial biomass in more proximal sites, e.g. as shown in this study PMID: 33268457. If histological or anatomical data are available, it would be informative to know whether any such gradient was observed in this cohort, or if the bacterial signal was uniformly low across all sampled regions. Even a brief mention of this consideration, or a statement about the availability of such data, would add valuable context to the findings.

We agree with the reviewer that such an analysis would be interesting. However, the large majority (811/940 cancers) of our data was based on the adenocarcinoma subtype, of distal origin (alveolar parenchyma), as expected in subjects who have never smoked. We had data on only 40 squamous cell carcinomas (proximal respiratory region), and we observed no significant differences between the two histological subtypes with either biomass (16S rRNA gene sequencing, P-value=0.54; RNAseq, P-value=0.11; WGS, P-value=0.94) or community composition (alpha diversity: 16S rRNA gene sequencing, P-value=0.14; RNAseq, P-value=0.54; WGS, P-value=0.23).

However, prompted by the reviewer's suggestion, we also explored data we had for a subset of individuals (n=553) on the anatomical location of the tumors spanning the upper, middle, and lower lobes, as well as both the left and right lungs. All associations with bacterial biomass and diversity in all datasets were not significant. Nonetheless, we thank the reviewer for helping us to go deeper into the analysis by anatomical locations:

Data	term	estimate	std.error	p.value	fd
WGS	LEFT_LOWER	Reference			
WGS	LEFT_UPPER	-0.200389562	0.1332484	0.13393250	1.0000000
WGS	RIGHT_LOWER	-0.078719325	0.1353114	0.56127400	1.0000000
WGS	RIGHT_MIDDLE	0.132928173	0.2269348	0.55859299	1.0000000
WGS	RIGHT_UPPER	0.189112790	0.1261580	0.13518929	1.0000000
RNA-seq	LEFT_LOWER	Reference			
RNA-seq	LEFT_UPPER	0.003322246	0.1073816	0.97538544	1.0000000
RNA-seq	RIGHT_LOWER	0.137529418	0.1188855	0.25033965	1.0000000
RNA-seq	RIGHT_MIDDLE	-0.341810154	0.2359604	0.15085156	1.0000000
RNA-seq	RIGHT_UPPER	0.012989765	0.1100989	0.90633858	1.0000000
16S	LEFT_LOWER	Reference			
16S	LEFT_UPPER	0.230431242	0.1217708	0.06109945	0.6720939
16S	RIGHT_LOWER	0.042316386	0.1407378	0.76423484	1.0000000
16S	RIGHT_MIDDLE	0.048785209	0.4879636	0.92054612	1.0000000
16S	RIGHT_UPPER	0.310497098	0.1232050	0.01317867	0.1581440

Response Table 1: Results of generalized linear model between \log_{10} bacterial reads per million and lobes of the lung, analyzing only adenocarcinomas and adjusted for study site.

experiment	term	estimate	std.error	p.value	fd
WGS	LEFT_LOWER	Reference			
WGS	LEFT_UPPER	0.14479079	0.09855249	0.14316998	1.0000000
WGS	RIGHT_LOWER	0.06924294	0.10107052	0.49398273	1.0000000
WGS	RIGHT_MIDDLE	0.30848144	0.17262203	0.07526624	0.8279286
WGS	RIGHT_UPPER	0.10771462	0.09472303	0.25667308	1.0000000
RNA-seq	LEFT_LOWER	Reference			
RNA-seq	LEFT_UPPER	0.29695132	0.14212347	0.03943312	0.4731974
RNA-seq	RIGHT_LOWER	0.13733626	0.15734930	0.38503826	1.0000000
RNA-seq	RIGHT_MIDDLE	-0.19844150	0.31230231	0.52673528	1.0000000
RNA-seq	RIGHT_UPPER	0.08417461	0.14571997	0.56491400	1.0000000
16S	LEFT_LOWER	Reference			
16S	LEFT_UPPER	0.12932368	0.12064424	0.28667690	1.0000000
16S	RIGHT_LOWER	0.15514894	0.14567858	0.28978553	1.0000000
16S	RIGHT_MIDDLE	0.31459716	0.43507077	0.47153957	1.0000000
16S	RIGHT_UPPER	0.10710938	0.11824582	0.36750485	1.0000000

Response Table 2: Results of generalized linear model between Shannon diversity and lobes of the lung, analyzing only adenocarcinomas and adjusted for study site.

Finally, a landmark methodological paper on use of different biospecimens for lung

microbiota analysis warrants citations in my view.
PMID: 31661299

We thank the reviewer for pointing out this omission – we have now included a reference to this manuscript (reference n. 33) in our Introduction where we discuss the discrepancies between BAL and lung tissue biospecimens:

Introduction

“In contrast, small-scale studies conducted on surgically removed tumor and normal lung tissue - which theoretically precludes contamination from the upper airways³³ - identified much lower proportions of upper airway bacteria^{34–38}.”

Reviewer #2 (Remarks to the Author):

In this impressive study, the authors seek to identify a lung cancer-associated microbiome signature by studying lung biopsy and blood samples from 940 patients. They use a combination of 16S, WGS and RNAseq to characterize the microbial communities of tumors and adjacent non-tumor and blood samples. Ultimately the study findings were negative- across platforms, alpha/beta diversity and individual genera did not associate with age, sex, ancestry, passive smoking, histology, stage, or survival. The study was sufficiently powered and the conclusions are believable. Generally, the methods are sound, and the authors make a reasonable attempt to address batch effects and background contamination, although the lack of negative controls for WGS and RNAseq data is suboptimal.

All said, this is an important contribution to the literature and in this case, a negative result is valuable knowledge for the scientific community.

We truly appreciate the reviewer’s positive assessment of our work.

I have some suggestions for further improving this study:

1. The authors used the simple but somewhat outdated ALDEx2 algorithm for differential abundance analyses. Can the authors confirm that using a more modern and widely accepted tool such as ANCOM-BC (<https://www.nature.com/articles/s41467-020-17041-7>) does not globally change the primary study conclusions?

We thank the reviewer for the important suggestion. We re-ran differential abundance analysis using ANCOM-BC and included these results in **Figure 3** and **Supplementary Figure 4**. Ultimately, we found no major differences between the results of ALDEx2 and ANCOM-BC differential abundance analyses. We have added this information in the Results and Methods:

Results:

“Comparing the tumor versus normal lung microbiome in 16S data, we identified few differentially abundant bacteria which were not significant after multiple testing correction (**Figure 3a, Supplementary Figure 4a, Supplementary Table 6**), and observed slightly decreased alpha diversity in tumor samples (mean diversity=1.9) compared to paired normal tissues (mean diversity=2.0, Wilcoxon $p=0.0016$, Figure 3b). Using RNA-seq, several bacterial genera were enriched in normal tissue compared with tumors (**Figure 3c, Supplementary Figure 4b, Supplementary Table 7**)...”

“...and several species of *Pseudomonas* and *Acinetobacter* were marginally enriched in tumor tissue versus normal lung tissue when analyzed using ANCOM-BC (**Supplementary Figure 4c,d**).”

Methods:

Differential Abundance

Differential abundance was analyzed using the ALDEx2⁹⁷ and the ANCOMBC⁹⁸ R packages. Bacterial taxa with prevalence less than 5% were discarded. For tumor versus normal differential abundance analysis, only subjects with paired tumor and normal lung tissue were included. For ANCOM-BC analysis, both study site and tumor-normal status were included in the differential abundance model to adjust for lingering batch effects.

2. Can the authors please justify their minimum read depth thresholds?

We had originally included a section describing our rationale for choosing read depth thresholds in the Methods section. However, we acknowledge that it could be easier for the reader to include this explanation within the Results. So, we have moved the section to the Results:

“For downstream statistical tests, RNA-seq samples with less than 500 reads were excluded to improve the reliability of associations. Due to the considerably lower read depth of 16S and WGS samples, this read cutoff was relaxed to 250 reads in 16S and 100 reads in WGS to preserve sample size. For intra-class correlation analyses (phylum-level relative abundances, alpha diversity, beta diversity), a cutoff of 250 reads was applied to all datasets to allow for valid comparisons.”

To verify that increasing the reads threshold would not change our results, we reran the major analyses with a more stringent cutoff: 1000 reads for RNAseq, 500 reads for 16S samples, and

500 reads for WGS samples. The results are largely unchanged (Reviewer Response Figure 1-3 below).

Reviewer Response Figure 1: Tumor-normal, clinical, and demographic associations with the microbiome. a) ANCOM-BC differential abundance results and b) comparison of Shannon alpha diversity between paired tumor and normal samples using genus-level 16S data. c) ANCOM-BC differential abundance results and d) comparison of Shannon alpha diversity between paired tumor and normal samples using genus-level RNA-seq data. e) Genus-level alpha diversity and richness associations in tumors with clinical features using a generalized linear model. RNA-seq, 16S, and WGS samples were rarefied to 1000, 500, and 500 bacterial reads respectively, and all associations were adjusted for study site. Stage I tumors, adenocarcinoma histology, and European (EUR)

ancestry serve as references. All tests are non-significant (FDR>0.05) after multiple testing correction. Error bars signify standard error.

Reviewer Response Figure 2: Survival associations with individual bacterial taxa. a) Cox proportional hazard model of ten-year survival with RNA-seq bacterial relative abundances including genera with minimum 50 reads in 10% of samples, b) 16S bacterial relative abundances including genera with minimum 10 reads in 10% of samples, and c) WGS bacterial relative abundances including genera with minimum 10 reads in 10% of samples. All analyses were stratified by study site, age at diagnosis (age>65 or ≤65), and stage (stage I or stages II-IV), and further adjusted by histology and age in ten year categories. All associations are not significant (FDR>0.05) after multiple testing correction. Error bars signify standard error.

Reviewer Response Figure 3: a) Overall survival association with RNA-seq tumor genus-level richness and b) Shannon alpha diversity, c) 16S tumor genus richness and d) Shannon alpha diversity, e) WGS tumor richness and f) Shannon alpha diversity, and g) WGS blood genus richness and h) Shannon alpha diversity. Cox models used continuous values for richness and diversity, which were then partitioned above or below the median within each study site for plotting. RNA-seq, 16S, and WGS samples were rarefied to 1000, 500, and 500 genus-level reads, respectively, for diversity and richness calculations.

3. Can the authors report effect sizes (e.g., \log_2FC and 95% Cis for enriched taxa)?

We have included new **Supplementary Tables 6-8** with the results of all differential abundance analyses, including log fold change and confidence intervals.

4. Can the authors please include read depth statistics (e.g., mean, range, std deviation of reads per sample)?

We have created a table in the main paper, **Table 2**, with read depth statistics for all samples. We thank the reviewer for this important suggestion.

5. The lack of negative controls and background correction for the WGS and RNAseq datasets is somewhat concerning. The reason for this is understandable however given that the data analyzed were not initially collected for metagenomic analysis. The authors reasonably compensate for this by subtracting reads from a previously curated list of established metagenomic contaminants. The authors state however that they “manually” removed taxa. Can the authors clarify whether they used a systematic computational approach to remove the same blacklisted taxa (Supp Table 7?) from every sample, or whether they indeed manually subtracted taxa on a sample-by-sample basis?

“Manually” in this context was meant to communicate that these taxa were not identified *de novo*, but instead removed based on a pre-compiled list of taxa. Taxa in this list were then removed in a systematic fashion. We have removed the word ‘manually’ to avoid any confusion.

6. The authors have provided only the submission reference numbers and not the accession numbers for their data, which are not accessible for review

We thank the reviewer to point out our omission. We have now provided the accession numbers for the data as follows:

“All microbiome data can be found in the following data archives: whole genome sequencing data, accession phs001697.v2.p1; 16S rRNA gene sequencing data, BioProject accession PRJNA1337178; RNA-seq data, accession phs003955.v1.p1.”

Reviewer #2 (Remarks on code availability):

I did not run the code because the intermediate files necessary to do so are unavailable on the GitHub repository. Upon inspection, the code appears clean and well annotated.

It would be helpful if the authors provided taxonomic count data on their GitHub repo or in supplemental data tables so the R code in the repo could be run to reproduce the results/figures without de-nova re-alignment of the raw sequencing reads.

We agree with the reviewer that the raw taxonomic count data needs to be available. We had reported these data in **Supplementary Tables 2 – 4**. These counts could be used to re-analyze the data and obtain the same results. All relevant metadata variables are also available in **Supplementary Table 1**.

Response to Reviewer 1

Reviewer 1 wrote:

The authors have responded nicely to my comments and in my view, to the comments of the other reviewer. I maintain my high level of appreciation for this in-depth work with overall neutral findings for the role of lung microbiome in lung cancer of never smokers.

The only minor comment I have pertains to the lung-blood correlation analysis, which was done at the Phylum level. I am afraid that this can create some spurious association and lead the reader to conclude that the significant but weak correlation for Firmicutes taxa may suggest lung origin of the detected taxa in the blood. There are stronger competing explanations, such as subclinical gut translocation or skin contamination/translocation of Firmicutes; unless the authors pursue a comparison at a lower taxonomic level (e.g. genera) and show some matching between lung and blood taxa, I recommend staying away from oversimplification for a plausible lung-origin of detectable DNA in the circulation, or at a minimum acknowledging the competing explanations in the Discussion.

As suggested by the Reviewer, we have reanalyzed the data at the genus level and found a strong correlation between blood and lung tissue for the genus *Staphylococcus*. We have now added this result to the Results section and the **Supplementary Figure 7c**. We are grateful to the Reviewer for this important suggestion. We have also modified the Discussion to include the following statement:

Of note, abundance of phyla Firmicutes and Proteobacteria as well as the abundance of genus Staphylococcus were correlated between tumor samples and paired blood. This suggests that these bacteria may migrate from lung tissue to blood, although translocation from the gut and/or contamination from the skin are always possible contributing factors. Nonetheless, we ultimately found no clinical associations with circulating bacteria.